# Policy Regularization on Globally Accessible States in Cross-Dynamics Reinforcement Learning

Zhenghai Xue [1]  Lang Feng [1]  Jiacheng Xu [1 2]  Kang Kang [2]  Xiang Wen [2]  Bo An [1 2]  Shuicheng YAN [2 3]

## Abstract

To learn from data collected in diverse dynamics, Imitation from Observation (IfO) methods leverage expert state trajectories based on the premise that recovering expert state distributions in other dynamics facilitates policy learning in the current one. However, Imitation Learning inherently imposes a performance upper bound of learned policies. Additionally, as the environment dynamics change, certain expert states may become inaccessible, rendering their distributions less valuable for imitation. To address this, we propose a novel framework that integrates reward maximization with IfO, employing $\mathcal{F}$-distance regularized policy optimization. This framework enforces constraints on globally accessible states—those with nonzero visitation frequency across all considered dynamics—mitigating the challenge posed by inaccessible states. By instantiating $\mathcal{F}$-distance in different ways, we derive two theoretical analysis and develop a practical algorithm called **A**ccessible **S**tate **O**riented Policy **R**egularization (ASOR). ASOR serves as a general add-on module that can be incorporated into various RL approaches, including offline RL and off-policy RL. Extensive experiments across multiple benchmarks demonstrate ASOR's effectiveness in enhancing state-of-the-art cross-domain policy transfer algorithms, significantly improving their performance.

## 1. Introduction

Imitation Learning (IL) and Reinforcement Learning (RL) (Sutton & Barto, 1998) facilitates large-scale policy optimization using datasets that are either static (Wu et al., 2019; Fujimoto et al., 2019) or continuously updated during training (Lillicrap et al., 2016; Haarnoja et al., 2018). When applied to non-stationary environments with evolving dynamics, obtaining effective policies necessitates extensive training datasets with sufficient dynamics coverage (Liu et al., 2022; Li et al., 2023). This challenge arises because optimal policies and state-action distributions are inherently dynamics-specific and do not generalize well. As a result, trajectory data collected under one dynamics cannot be directly utilized for policy learning in another.

To address inefficient data exploitation, IfO algorithms (Wu et al., 2019; Torabi et al., 2018b; Jiang et al., 2020) propose a dynamics-agnostic approach that enables learning from data with varying dynamics. It imitates stationary state distributions of expert policies and relies on the idea that expert state distributions are similar across different dynamics (Gangwani & Peng, 2020; Desai et al., 2020; Radosavovic et al., 2021). Building on this idea, Xue et al. (2023a) further shows that the learning policy can achieve strong cross-dynamics performance if it recovers the expert state distribution in at least one of the dynamics. Instead, we highlight a key limitation of this idea: the similarity of state distributions does not always hold, particularly in tasks where state accessibility varies with environmental dynamics. In such cases, certain expert states become inaccessible as the environment changes. For example, an autonomous vehicle may safely navigate intersections at high speed under low traffic densities, but will face a high risk of collisions in dense traffic. Consequently, states representing "safe driving at high speed" become inaccessible in certain dynamics, leading to distinct stationary state distributions. In such scenarios, expert trajectories with dynamics shift can be misleading.

To deal with the issue of distinct state accessibility, expert states that are not visited in some dynamics should be excluded during training. We define *globally accessible states* which maintains the same accessibility across different dynamics. By restricting imitation to these states, the policy can still leverage expert state trajectories from various dynamics while avoiding misleading information from inaccessible states. Meanwhile, IL cannot be naturally integrated with datasets containing reward signals and requires

---

[1]Nanyang Technological University, Singapore [2]Skywork AI [3]National University of Singapore. Correspondence to: Bo An <boan@ntu.edu.sg>.

*Proceedings of the 42nd International Conference on Machine Learning*, Vancouver, Canada. PMLR 267, 2025. Copyright 2025 by the author(s).

demonstrations to be optimal. To overcome this limitation, we propose a policy regularization method that incorporates imitation as constraint while optimizing the reward maximization objective in RL. Specifically, the constraint ensures that the $\mathcal{F}$-distance (Arora et al., 2017) between the accessible state distributions of the current and expert policies remains upper-bounded. This approach offers two key advantages: By instantiating the $\mathcal{F}$-distance with JS divergence and network distance, we establish lower-bound performance guarantees for policy regularization under dynamics shift; By instantiating the $\mathcal{F}$-distance with a GAN-like distance measure, we transform policy regularization into a practical reward augmentation algorithm which can be a general add-on module to existing cross-dynamics RL algorithms (Chen et al., 2021; Luo et al., 2022).

In empirical evaluations, we access the proposed algorithm across various environments, including Minigrid (Chevalier-Boisvert et al., 2023), the simulated robotics environment MuJoCo (Todorov et al., 2012), the simulated autonomous-driving environment MetaDrive (Li et al., 2023), and a large-scale fall guys-like game environment. The proposed algorithm exhibits superior performance when integrated with multiple state-of-the-art algorithms. Our contributions can be summarized as follows: 1) We identify a common limitation of existing IfO methods under dynamics shift and propose state distribution imitation restricted to globally accessible states; 2) We design an $\mathcal{F}$-distance regularized policy optimization framework that combines expert imitation with reward maximization; 3) By instantiating the $\mathcal{F}$-distance in different ways, we conduct theoretical analyses and introduce a practical algorithm, both validating the effectiveness of policy regularization on globally accessible states.

## 2. Backgroud

### 2.1. Preliminaries

To model a set of decision-making tasks with different environment dynamics, we consider the Hidden Parameter Markov Decision Process (HiP-MDP) (Doshi-Velez & Konidaris, 2016) defined by a tuple $(\mathcal{S}, \mathcal{A}, \Theta, T, r, \gamma, \rho_0)$, where $\mathcal{S}$ is the state space and $\mathcal{A}$ is the bounded action space with actions $a \in [-1, 1]$. $\Theta$ is the space of hidden parameters. $T_\theta(s'|s, a)$ is the transition function conditioned on $(s, a)$, as well as a hidden parameter $\theta$ sampled from $\Theta$. $r(s, a, s')$ is the environment reward function. By taking all $s, a, s'$ into account, the reward function inherently includes the transition information and does not change in different dynamics. We also assume $r(s, a, s')$ w.r.t. the action $a$ is $\lambda$-Lipschitz. Discussions on these Lipschitz properties can be found in Appendix A.3. $s'$ is termed as *accessible* from

$s$ under dynamics $T^1$ if $\sum_{a \in \mathcal{A}} T(s'|s, a) > 0$. $\gamma \in (0, 1)$ is the discount factor and $\rho_0(s)$ is the initial state distribution.

Policy optimization under dynamics shift aims at finding the optimal policy that maximizes the expected return under all possible $\theta \in \Theta$: $\pi^* = \arg\max_\pi \eta(\pi) = \mathbb{E}_\theta \mathbb{E}_{\pi, T_\theta} [\sum_{t=0}^\infty \gamma^t r(s_t, a_t, s_{t+1})]$, where the expectation is under $s_0 \sim \rho_0$, $a_t \sim \pi(\cdot|s_t)$, and $s_{t+1} \sim T_\theta(\cdot|s_t, a_t)$. The Q-value $Q_T^\pi(s, a)$ denotes the expected return after taking action $a$ at state $s$: $Q_T^\pi(s, a) = E_{\pi, T} [\sum_{t=0}^\infty \gamma^t r(s_t, a_t, s_{t+1})|s_0 = s, a_0 = a]$. The value function is defined as $V_T^\pi(s) = \mathbb{E}_{a \sim \pi(\cdot|s)} Q_T^\pi(s, a)$. The optimal policy $\pi^*$ under $T$ is defined as $\pi_T^* = \arg\max_\pi \mathbb{E}_{s \sim \rho_0} V_T^\pi(s)$. We also intensely use the stationary state distribution (also referred to as the state occupation function) $d_T^\pi(s) = (1 - \gamma) \sum_{t=0}^\infty \gamma^t p(s_t = s \mid \pi, T)$. The stationary state distribution under the optimal policy is denoted as $d_T^*(s)$, which is the shorthand for $d_T^{\pi_T^*}(s)$. $d_T^*(s)$ will be briefly termed as optimal state distribution in the rest of this paper.

### 2.2. Related Work

**Cross-domain Policy Transfer** Cross-domain policy transfer (Niu et al., 2024) focuses on training policies in source domains and testing them in the target domain. In this paper, we focus on a related problem of efficient training in multiple source domains. The resulting algorithm can be combined with any of the following cross-domain policy transfer algorithms to improve the test-time performance. In online RL, VariBAD (Zintgraf et al., 2020) trains a context encoder with variational inference and trajectory likelihood maximization. CaDM (Lee et al., 2020) and ESCP (Luo et al., 2022) construct auxiliary tasks including next state prediction and contrastive learning to train the encoders. Instead of relying on context encoders, DARC (Eysenbach et al., 2021) makes domain adaptation by assigning higher rewards on samples that are more likely to happen in the target environment. Encoder-based (Chen et al., 2021) and reward-based (Liu et al., 2022) policy transfer algorithms are also effective in offline policy adaptation and have been extended to offline-to-online tasks (Niu et al., 2022; 2023). VGDF (Xu et al., 2023) use ensembled value estimations to perform prioritized Q-value updates, which can be applied in both online and offline settings. SRPO (Xue et al., 2023a) focus on a similar setting of efficient data usage with this paper, but is based on a strong assumption of universal identical state accessibility. We demonstrate that such an assumption will not hold in many tasks and a more delicate characterization of state accessibility will lead to better theoretical and empirical results.

---

[1]$T$ without subscript $\theta$ refers to the transition function under any of the hidden parameter $\theta$.

**Imitation Learning from Observations** Imitation Learning from Observation (IfO) approaches obviate the need of imitating expert actions and is suitable for tasks where action demonstrations may be unavailable. BCO (Wu et al., 2019) and GAIfO (Torabi et al., 2018b) are two natural modifications of traditional Imitation Learning (IL) methods (Ho & Ermon, 2016) with the idea of IfO. IfO has also been found promising when the demonstrations are collected from several environments with different dynamics. Usually an inverse dynamics model is first trained with samples from the target environment by supervised learning (Wu et al., 2019; Radosavovic et al., 2021), variational inference (Liu et al., 2020), or distribution matching (Desai et al., 2020). It is then used to recover the adapted actions in samples from the source environment. The recovered samples can be used to update policies with action discrepancy loss (Gangwani & Peng, 2020; Radosavovic et al., 2021; Liu et al., 2020). To our best knowledge, only HIDIL (Jiang et al., 2020) considered state distribution mismatch across different dynamics, where policies are allowed to take extra steps to reach the next state specified in the expert demonstration. We consider in this paper a more general setting where states in expert demonstrations may even be inaccessible.

## 3. Accessible States Oriented Policy Learning

In this section, we first provide a motivating example in Sec. 3.1, where expert state trajectories can mislead the learning policy in cross-dynamics training due to different state accessibility. Based on the globally accessible states defined in Sec. 3.2, we then propose a general approach of $\mathcal{F}$-distance regularized policy optimization. In Sec. 3.3 and Sec. 3.4, we propose two instantiations of the $\mathcal{F}$-distance and respectively provide infinite-sample and finite-sample performance analyses for the policy regularization method. In Sec. 3.5, we propose a GAN-like instantiation of the $\mathcal{F}$-distance, giving rise to the practical algorithm.

### 3.1. Motivating Example

Previous IfO approaches imitate expert policies through state distribution matching (Torabi et al., 2018a; Desai et al., 2020; Jiang et al., 2020) for cross-dynamics policy training. Fig. 1 demonstrates a lava world task with dynamics shift where expert state distributions are no longer reliable. Considering a three dimensional state space including agent row, agent column, and a 0-1 variable indicating whether there is an accessible lava block within one step reach of the agent. The agent starts from the blue grid and targets at the green grid with positive reward. It also receives a small negative reward on each step. The red grid stands for the dangerous lava area which ends the trajectory on agent entering. One lava block is fixed at Row 1, while the other may appear at Row 2, 3, 4, and 5. Fig. 1 demonstrates two examples where

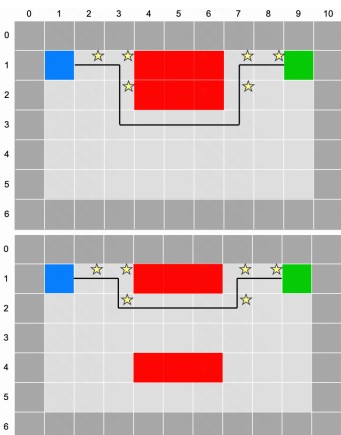

*Figure 1.* Lava world example with dynamics shift.

the movable lava block is at Row 2 and Row 4. Taking state (1,3,0) as an example, the same action of "moving down" gives rise to different next state distributions due to distinct lava positions, leading to environment dynamics shift.

The state trajectories of the optimal policies on two example lava environments are plotted with black lines. The optimal state distributions are different under distinct environment dynamics. For example, state (3,3,0) has non-zero probability under $d^*(s)$ in Fig. 1 (above), but cannot be visited by the optimal policy in Fig. 1 (bottom). Existing state-only policy transfer algorithms will therefore not be suitable for such seemingly simple task, as demonstrated by the empirical results in Appendix C.2. The main cause of this distribution difference is the *break of accessibility*. State (4,2) is accessible from (3,2) in Fig. 1 (lower), but is inaccessible in Fig. 1 (upper). The inaccessible states will certainly have zero visitation probability and make the optimal state distribution different. Such break of accessibility can also happen in various real-world tasks, as discussed in Appendix B.

### 3.2. $\mathcal{F}$-distance Regularized Policy Optimization with Accessible State Distribution

Motivated by examples in Sec. 3.1, we propose to ignore inaccessible states and focus on states that are accessible under all possible dynamics. We term the latter as *globally accessible states* with the following formal definition.

**Definition 3.1** (Globally Accessible States)**.** In an HiP-MDP $(\mathcal{S}, \mathcal{A}, \Theta, T, r, \gamma, \rho_0)$, a state $s \in \mathcal{S}$ is called an *globally accessible state* if for all $\theta \in \Theta$, there exist a policy $\pi$, such that $d_{T_\theta}^\pi(s) > 0$. $S^+ \subseteq S$ is denoted as the set of globally accessible states. The *accessible state distribution* of policy $\pi$ under dynamics $T$ is defined as $d_T^{\pi,+}(s) = (1 - \gamma) \sum_{t=0}^{\infty} \gamma^t p\left(s_t = s, s_t \in S^+ | \pi, T\right) / Z(\pi)$, where $Z(\pi) = \sum_{t=0}^{\infty} \gamma^t p\left(s_t \in S^+ | \pi, T\right)$ is the normalizing term.

In the lava world example in Fig. 1, the intersection of

optimal state trajectories and globally accessible states is marked with yellow stars. These states are more likely to be visited by the expert policies across both dynamics, effectively filtering out misleading states that are optimal in one dynamics and suboptimal in others. Consequently, expert state distributions can be safely imitated on globally accessible states under dynamics shifts.

Meanwhile, IL relies on access to expert trajectories, limiting its applicability in broader scenarios. In a more general RL setting, where the dataset consists of trajectories with mixed policy performance and reward labels, policy optimization to maximize environment reward is often more preferable. To simultaneously leverage the expert state distribution, we propose to regularize the training policy to generate a stationary state distribution that closely aligns with the expert accessible state distribution $d_T^{*,+}(s)$. The regularization is exerted with an upper-bound constraint on the $\mathcal{F}$-distance.

**Definition 3.2** ($\mathcal{F}$-distance, Definition 2 in (Arora et al., 2017)). Let $\mathcal{F}$ be a class of functions from $\mathbb{R}^d$ to $[0,1]$ such that if $f \in \mathcal{F}, 1 - f \in \mathcal{F}$. Let $\phi$ be a concave measuring function. Then the $\mathcal{F}$-divergence with respect to $\phi$ between two distributions $\mu$ and $\nu$ is defined as

$$d_{\mathcal{F},\phi}(\mu,\nu) = \sup_{\omega \in \mathcal{F}} \mathbb{E}_{x \sim \mu}[\phi(\omega(x))] + \mathbb{E}_{x \sim \nu}[\phi(1 - \omega(x))] - C(\phi),$$

where $C(\phi)$ is irrelevant to $\mathcal{F}, \mu$, and $\nu$.

The resulting regularized policy optimization problem is formulated as follows:

$$\max_{\pi} \mathbb{E}_{\theta,\tau_\pi} \sum_{t=0}^{\infty} \gamma^t r\left(s_t, a_t, s_{t+1}\right)$$
$$\text{s.t. } \max_T d_{\mathcal{F},\phi}\left(d_T^\pi(\cdot), d_{T_0}^{*,+}(\cdot)\right) < \varepsilon, \tag{1}$$

where $T_0$ is an arbitrary environment dynamics. In the following, we will demonstrate the effectiveness of this policy regularization framework both theoretically and empirically.

### 3.3. Infinite Sample Analysis with JS Divergence Instantiation

In Sec. 3.3 and Sec. 3.4, we conduct theoretical analysis on how the policy regularization method in Eq. (1) influences policy performance under dynamics shift. We start with the definition of $M$-$R_s$ accessible MDPs, which formally characterizes the required accessibility property of MDPs so that expert state distributions can be imitated in a dynamics-agnostic approach.

**Definition 3.3.** Consider MDPs $\mathcal{M}_1 = (\mathcal{S}, \mathcal{A}, T_1, r, \gamma, \rho_0)$ and $\mathcal{M}_2 = (\mathcal{S}, \mathcal{A}, T_2, r, \gamma, \rho_0)$. If for all transitions $(s, a, s')$ with $T_1(s'|a, s) > 0$, there exists states $s_0, s_1, \cdots, s_N$ and actions $a_0, a_1, \cdots, a_{N-1}$ such that

$N \leqslant M, s_0 = s, s_N = s', \prod_{n=1}^{N} T_2(s_n|s_{n-1}, a_{n-1}) > 0$, and

$$\left| \sum_{n=1}^{N-1} \gamma^{n-1} r(s_n, a_n, s_{n+1}) + (1 - \gamma^{N-1}) V_{T_1}^*(s_0) \right| \leqslant R_s,$$

$\mathcal{M}_2$ is referred to as $M$-$R_s$ accessible from $\mathcal{M}_1$.

In this definition, $M$ is the number of extra steps required in $\mathcal{M}_2$ to reach the state $s'$ from $s$, compared with in $\mathcal{M}_1$. $R_s$ constrains the reward discrepancy in these extra steps. One special case is when $\mathcal{M}_2$ and $\mathcal{M}_1$ are 1-0 accessible from each other, all states between $(s, s')$ will have the same state accessibility. It is identical to the property of "homomorphous MDPs" (Xue et al., 2023a), based on which a theorem about identical optimal state distribution can be proved. Most of the previous approaches in IfO is built upon such assumption of 1-0-accessible MDPs, which is an over-simplification in many tasks. For example, the Minigrid environment in Sec. 3.1 contains 3-0.03 accessible MDPs. Instead, our derivations are based on the milder assumption on $M$-$R_s$ accessible MDPs that fits for a broader range of practical scenarios.

We first consider the case where policy regularization is performed with infinite samples, so that the constraint on $\mathcal{F}$-distance is followed during both training and validation. We utilize JS divergence with the following instantiation of the $\mathcal{F}$-distance.

**Proposition 3.4.** *When $\phi(t) = \log(t)$ and $\mathcal{F} = \{$ all functions from $\mathbb{R}^d$ to $[0,1]\}$, $d_{\mathcal{F},\phi}$ is the JS divergence.*

We show that the learning policy $\hat{\pi}$ will have a performance lower-bound given a bounded JS-divergence with the optimal accessible state distribution.

**Theorem 3.5.** *Consider the HiP-MDP $(\mathcal{S}, \mathcal{A}, \Theta, T, r, \gamma, \rho_0)$ with its MDPs $M$-$R_s$ accessible from each other. For all policy $\hat{\pi}$, if there exists one certain dynamics $T_0$ such that $\max_{\theta \in \Theta} D_{\text{JS}}(d_{T_\theta}^{\hat{\pi}}(\cdot) \| d_{T_0}^{*,+}(\cdot)) \leqslant \varepsilon$, we have*

$$\eta(\hat{\pi}) \geqslant \max_{\theta} \eta(\pi_{T_\theta}^*) - \frac{2R_s + 6\lambda + 2R_{\max}\varepsilon}{1 - \gamma}. \tag{2}$$

Previous approaches (Xu et al., 2023; Janner et al., 2019; Xue et al., 2023b) also provide policy performance lower-bounds, but these bounds have *quadratic* dependencies on the effective planning horizon $\frac{1}{1-\gamma}$. By accessible state-based policy regularization, we obtain a tighter discrepancy bound with *linear* dependency on the effective horizon.

### 3.4. Finite Sample Analysis with Network Distance Instantiation

In this subsection, we derive a performance lower-bound of $\hat{\pi}$ if it is regularized with finite samples from the optimal accessible state distribution. Due to the limited generalization

ability of JS distance (Xu et al., 2020), we characterize the regularization error in Eq. (1) with the network distance.

**Proposition 3.6** (Neural network distance (Arora et al., 2017))**.** *When $\phi(t) = t$ and $\mathcal{F}$ is the set of neural networks, $\mathcal{F}$-distance is the network distance: $d_{\mathcal{F}}(\mu, \nu) = \sup_{F \in \mathcal{F}} \{ \mathbb{E}_{s \sim \mu}[F(s)] - \mathbb{E}_{s \sim \nu}[F(s)] \}$.*

Given $m$ samples and the network distance bounded by $\varepsilon_{\mathcal{F}}$, we analyze the generalization ability of the policy regularization in Eq. (1) with the following theorem.

**Theorem 3.7.** *Consider the HiP-MDP $(\mathcal{S}, \mathcal{A}, \Theta, T, r, \gamma, \rho_0)$ with its MDPs $M$-$R_s$ accessible from each other. Given $\{s^{(i)}\}_{i=1}^m$ sampled from $d_{T_0}^{+,*}$, for policy $\hat{\pi}$ regularized by $\hat{d}_{T_0}^{+,*}$ with the constraint $\max_{\theta \in \Theta} d_{\mathcal{F}}(\hat{d}_{T_\theta}^{\hat{\pi}}, \hat{d}_{T_0}^{+,*}) < \varepsilon_{\mathcal{F}}$, we have[2]*

$$\eta(\hat{\pi}) \geqslant \max_\theta \eta(\pi_{T_\theta}^*) - \frac{2R_s + 8\lambda}{1 - \gamma} - \mathcal{O}(\frac{1/\sqrt{m} + \varepsilon_{\mathcal{F}}}{1 - \gamma}) \tag{3}$$

*with probability at least $1 - \delta$, where $\hat{d}_T^{\hat{\pi}}$ and $\hat{d}_{T_0}^{+,*}$ are the empirical version of distributions $d_T^{\hat{\pi}}$ and $d_{T_0}^{+,*}$ on $\{s^{(i)}\}_{i=1}^m$.*

Thm. 3.7 shows that with finite samples, the policy regularization still leads to a tight performance lower-bound with linear horizon dependency. The lower-bound is stronger than the sample complexity analysis of Behavior Cloning with quadratic horizon dependency (Xu et al., 2020) and has the same horizon dependency with GAIL (Ho & Ermon, 2016).

### 3.5. Practical Algorithm with GAN-like Objective Function

In this subsection, we design practical algorithms to solve the policy regularization problem in Eq. (1). The first challenge is to estimate the value of a certain instantiation of the $\mathcal{F}$-distance. With the following proposition, we manage to do so with a GAN-like optimization process.

**Proposition 3.8.** *When $\mathcal{F}$ is a set of neural networks and $\phi(t) = \log t$, the $\mathcal{F}$-distance $d_{\mathcal{F},\phi}(\mu, \nu)$ is equivalent to the discriminator's objective function in a GAN (Goodfellow et al., 2014), where $\mu$ is the real data distribution $\hat{\mathcal{D}}_{real}$ and $\nu$ is the generated data distribution $\hat{\mathcal{D}}_G$.*

According to the proposition, $d_{\mathcal{F},\phi}\left(d_T^\pi(\cdot), d_{T_0}^{*,+}(\cdot)\right)$ in Eq. (1) can be obtained by training a GAN discriminator to classify two datasets, one sampled from $d_T^\pi(\cdot)$ and the other sampled from $d_{T_0}^{*,+}(\cdot)$. While $d_T^\pi(\cdot)$ can be related to data newly collected in the replay buffer (Liu et al., 2021; Sinha et al., 2022), sampling from $d_{T_0}^{*,+}(\cdot)$ is still challenging. We introduce the binary observation state $\mathcal{O}_t$ with $\mathcal{O}_t = 1$ denoting $s_t$ is the optimal state at timestep $t$ (Levine, 2018).

---

[2]The $\mathcal{O}$ notation omits constants irrelevant to $m, \varepsilon_F$, and $\gamma$. The full version is in Thm. A.4.

**Algorithm 1** The workflow of ASOR on top of ESCP (Luo et al., 2022).

---
1: **Input:** Training MDPs $\{\mathcal{M}_0, \cdots, \mathcal{M}_{n-1}\}$; Context encoder $\phi$; Policy network $\pi$; Value network $V$; Discriminator Network $\omega$; Rollout horizon $H$; State partition ratio $\rho_1, \rho_2$; Regularization coefficient $\lambda$; Replay Buffer $\mathcal{R}$.
2: **for** $step = 0, 1, 2, \ldots$ **do**
3:     Sample MDP $\mathcal{M}_i$ from $\{\mathcal{M}_0, \mathcal{M}_1, \cdots, \mathcal{M}_{n-1}\}$.
4:     **for** $t = 1, 2, \ldots, H$ **do**
5:         Sample $z_t$ from $\phi(z \mid s_t, a_{t-1}, z_{t-1})$ and then sample $a_t$ from $\pi(a \mid s_t, z_t)$, as in ESCP.
6:         Rollout and get transition data $(s_{t+1}, r_t, d_{t+1}, s_t, a_t, z_t)$ from $\mathcal{M}_i$; Add the data to the replay buffer $\mathcal{R}$.
7:     Sample a batch $D_{\text{batch}}$; Add $\rho_1\rho_2|D_{\text{batch}}|$ states with top $\rho_1$ portion of high values and $\rho_2$ portion of high proxy visitation counts to $D_P$; Add other states to $D_Q$.
8:     Train $\omega$ as a GAN discriminator, regarding $D_P$ as the real dataset and $D_Q$ as the generated dataset.
9:     For one-step transition in $D_{\text{batch}}$, update $r_t$ with $r_t + \lambda \log \omega(s_t)$.
10:    Use the updated $D_{\text{batch}}$ to update $\phi, \pi$, and $V$.

---

$d_{T_0}^{*,+}(s_t)$ can therefore be written as $d_{T_0}^{\pi,+}(s|\mathcal{O}_{0:\infty})$, which is the distribution of the states generated by a certain $\pi$, given these states are optimal. With the Bayes' rule, we have

$$
\begin{aligned}
\frac{d_{T_0}^{*,+}(s)}{d_T^\pi(s)} &= \frac{d_{T_0}^{\pi,+}(s|\mathcal{O}_{0:\infty})}{d_T^\pi(s)} \\
&= \frac{p(\mathcal{O}_{0:\infty}|s, \pi, T_0)d_{T_0}^{\pi,+}(s)}{p(\mathcal{O}_{0:\infty}|\pi, T_0)} \cdot \frac{1}{d_T^\pi(s)} \\
&= \frac{p(\mathcal{O}_{0:\infty}|s, \pi, T_0)d_T^\pi(s)}{p(\mathcal{O}_{0:\infty}|\pi, T_0)} \cdot \frac{1}{d_T^\pi(s)} \cdot \frac{d_{T_0}^{\pi,+}(s)}{d_T^\pi(s)} \\
&= \frac{d_{T_0}^*(s)}{d_T^\pi(s)} \cdot \frac{d_{T_0}^{\pi,+}(s)}{d_T^\pi(s)}, \tag{4}
\end{aligned}
$$

where the last equation is also obtained with the Bayes' rule. According to Xue et al. (2023a), state $s$ will be more likely to be sampled from $d_{T_0}^*(\cdot)$ than from $d_T^\pi(\cdot)$ if it has a higher state value $V(s)$ than average. Meanwhile, $d_{T_0}^{\pi,+}(s)$ will be higher if $s$ falls in the set of globally accessible states $S^+$ and is most likely to be visited by various policies under dynamics shift. Therefore, any of the pseudo-count approaches of visitation frequency can be used to measure whether $s$ is more likely to be sampled from $d_{T_0}^{\pi,+}(s)$. In simulated environments with small observation spaces (Sec. 4.1 and 4.2), disagreement in next state predictions of ensembled environment models has shown to be a good proxy of visitation frequency (Yu et al., 2020). In large-scale real-world

tasks (Sec. 4.3), next state predictions can be unreliable, so we adopt Random Network Distillation (RND) (Burda et al., 2019) and use the error of predicting a random mapping as the proxy visitation measure.

With the trained discriminator $\omega^*(x)$ in the $\mathcal{F}$-distance, Eq. (1) can be transformed into an unconstrained optimization problem with the following Lagrangian to maximize[3]:

$$\mathbb{E}_{\theta, \pi, T_\theta} \left[ \sum_{t=0}^\infty \gamma^t \left( r\left(s_t, a_t, s_{t+1}\right) + \lambda \log \omega^*(s_t) \right) \right] + \frac{\lambda \varepsilon}{1 - \gamma}, \tag{5}$$

where $\lambda > 0$ is the Lagrangian Multiplier. The only difference between Eq. (5) and the standard RL objective is that $\log \omega^*(s_t)$ is augmented to the environment reward $r(s_t, a_t, s_{t+1})$. Therefore, the proposed approach can work as an add-on module to a wide range of RL algorithms with reward augmentation.

Summarizing previous derivations, we obtain a practical reward augmentation algorithm termed as ASOR (**A**ccessible **S**tate **O**riented **R**egularization) for policy optimization under dynamics shift. We select the ESCP (Luo et al., 2022) algorithm, which is one of the SOTA algorithms in online cross-dynamics policy training, as the example base algorithm. The detailed procedure of ESCP+ASOR is shown in Alg. 1. After the environment rollout and obtaining the replay buffer (line 6), we sample a batch of data from the buffer, obtain a portion of $\rho_1 \rho_2$ states with higher values and proxy visitation counts, and add them to the dataset $\mathcal{D}_P$. Other states are added to $\mathcal{D}_Q$. Then a discriminator network $\omega$ is trained (line 8) to estimate the logarithm of the optimal discriminator output $\lambda \log \omega^*(s)$, which is added to the reward $r_t$ (line 9). The effects of hyperparameters $\rho_1$, $\rho_2$, and $\lambda$ are investigated in Sec. 4.1. The procedure of the offline algorithm MAPLE (Lee et al., 2020)+ASOR is similar to ESCP+ASOR, where the datasets are built with data from the offline dataset rather than the replay buffer.

## 4. Experiments

In this section, we conduct experiments to investigate the following questions: (1) Can ASOR efficiently learn from data with dynamics shift and outperform current state-of-the-art algorithms? (2) Is ASOR general enough when applied to different styles of training environments, various sources of environment dynamics shift, and when combined with distinct algorithm setup? (3) How does each component of ASOR (e.g., the reward augmentation and the pseudo-count of state visitations) and its hyperparameters perform in practice? To answer questions (1)(2), we construct cross-dynamics training environments based on tasks including Minigrid (Chevalier-Boisvert et al., 2023)[4], D4RL (Fu et al.,

2020), MuJoCo (Todorov et al., 2012), and a Fall Guys-like Battle Royal Game. Dynamics shift in these environments comes from changes in navigation maps, evolvements of environment parameters, and different layouts of obstacles. To train RL policies in these environments, ASOR is implemented on top of algorithms including PPO (Schulman et al., 2017), MAPLE (Chen et al., 2021), and ESCP (Luo et al., 2022), which are all state-of-the-art approaches in the corresponding field. To answer question (3), we visualize how the learned discriminator and the pseudo state count behave in different environments. Moreover, ablation studies are conducted to examine the role of the discriminator and the influence of hyperparameters. Detailed descriptions of baseline algorithms are in Appendix C.1.

### 4.1. Results in Offline RL Benchmarks

For offline RL benchmarks, we collect the static dataset from environments with three different environment dynamics in the format of D4RL (Fu et al., 2020). Specifically, data from the original MuJoCo environments, environments with 3 times larger body mass, and environments with 10 times higher medium density are included. For baseline algorithms, we inlude IfO algorithms BCO (Torabi et al., 2018a) and SOIL (Radosavovic et al., 2021), standard offline RL algorithms CQL (Kumar et al., 2020) and MOPO (Yu et al., 2020), offline cross-domain policy transfer algorithms MAPLE (Chen et al., 2021), MAPLE+DARA (Liu et al., 2022), and MAPLE+SRPO (Xue et al., 2023a).

The comparative results are exhibited in Tab. 1. IfO approaches have the worst performance because they ignore the reward information (BCO) or cannot safely exploit the offline dataset (SOIL). Without the ability of cross-domain policy learning, CQL and MOPO cannot learn from data with dynamics shift and show inferior performances. Cross-domain policy transfer algorithms MAPLE, MAPLE+DARA, and MAPLE+SRPO show reasonable performance enhancement, while our MAPLE+ASOR algorithm leads to the highest performance. This highlights the effectiveness of policy regularization on accessible states. We discuss the conceptual advantages of ASOR compared with DARA and SRPO in Appendix A.4.

The results of ablation studies are shown in Tab. 2. They show that a larger value of reward augmentation coefficient $\lambda$ can give rise to performance increase. Meanwhile, using fixed values of $\lambda$, $\rho_1$ and $\rho_2$ will not lead to a large drop of performance scores, so ASOR is robust to hyperparameter changes. ASOR's will have degraded performance if training datasets $D_P$ and $D_Q$ are improperly constructed, e.g., with random data partition, or without considering state values and visitation counts, i.e., with fixed $\rho_1 = 0$ or fixed $\rho_2 = 0$.

---

[3]The derivations are in Appendix A.1.
[4]We leave the results in Appendix C.2.

*Table 1.* Results of offline experiments on MuJoCo tasks. Numbers before $\pm$ are scores normalized according to D4RL (Fu et al., 2020) and averaged across trials with four different seeds. Numbers after $\pm$ are normalized standard deviations. ME, M, MR and R correspond to the medium-expert, expert, medium-replay and random dataset, respectively.

| | BCO | SOIL | CQL | MOPO | MAPLE | MAPLE +DARA | MAPLE +SRPO | MAPLE +ASOR |
|---|---|---|---|---|---|---|---|---|
| Walker2d-ME | 0.25±0.04 | 0.14±0.08 | **0.63**±0.13 | 0.06±0.05 | 0.14±0.08 | 0.31±0.02 | 0.22±0.07 | 0.29±0.12 |
| Walker2d-M | 0.17±0.07 | 0.16±0.01 | **0.75**±0.02 | 0.15±0.22 | 0.41±0.19 | 0.46±0.10 | 0.32±0.17 | 0.49±0.04 |
| Walker2d-MR | 0.01±0.00 | 0.04±0.01 | 0.06±0.00 | -0.00±0.00 | 0.13±0.01 | 0.12±0.00 | 0.13±0.01 | **0.14**±0.01 |
| Walker2d-R | 0.00±0.00 | 0.00±0.00 | 0.00±0.00 | -0.00±0.00 | **0.22**±0.00 | 0.16±0.01 | 0.22±0.00 | 0.22±0.00 |
| Hopper-ME | 0.08±0.02 | 0.01±0.00 | 0.20±0.07 | 0.01±0.00 | 0.45±0.07 | 0.49±0.01 | 0.43±0.06 | **0.51**±0.06 |
| Hopper-M | 0.00±0.00 | 0.08±0.00 | 0.29±0.06 | 0.01±0.00 | 0.38±0.09 | 0.26±0.02 | 0.48±0.04 | **0.71**±0.14 |
| Hopper-MR | 0.00±0.00 | 0.00±0.00 | 0.08±0.00 | 0.01±0.01 | 0.55±0.17 | 0.75±0.10 | 0.73±0.16 | **0.76**±0.08 |
| Hopper-R | 0.00±0.00 | 0.00±0.00 | 0.10±0.00 | 0.01±0.00 | 0.12±0.00 | 0.12±0.00 | 0.25±0.08 | **0.32**±0.00 |
| HalfCheetah-ME | 0.43±0.00 | 0.00±0.00 | 0.03±0.04 | -0.03±0.00 | 0.53±0.07 | 0.39±0.00 | 0.58±0.04 | **0.61**±0.02 |
| HalfCheetah-M | 0.14±0.02 | 0.39±0.00 | 0.42±0.01 | 0.36±0.27 | 0.61±0.01 | **0.66**±0.03 | 0.62±0.00 | 0.62±0.01 |
| HalfCheetah-MR | 0.16±0.01 | 0.25±0.00 | 0.46±0.00 | -0.03±0.00 | 0.52±0.01 | 0.53±0.02 | 0.54±0.00 | **0.56**±0.01 |
| HalfCheetah-R | 0.14±0.01 | 0.35±0.01 | -0.01±0.01 | -0.03±0.00 | 0.20±0.02 | 0.19±0.01 | **0.22**±0.00 | 0.21±0.00 |
| Average | 0.11 | 0.11 | 0.25 | 0.04 | 0.36 | 0.37 | 0.40 | **0.45** |

*Table 2.* Results of ablation studies in Offline MuJoCo tasks. The scores are averaged on each environment with different expert levels.

| | Fixed $\lambda = 0.1$ | Fixed $\lambda = 0.3$ | Random partition | Fixed $\rho_1 = 0$ | Fixed $\rho_2 = 0$ | Fixed $\rho_1, \rho_2 = 0.5$ | Fixed $\rho_1, \rho_2 = 0.3$ | MAPLE +ASOR |
|---|---|---|---|---|---|---|---|---|
| Walker2d | 0.22 | 0.25 | 0.26 | 0.22 | 0.29 | **0.30** | 0.26 | 0.28 |
| Hopper | 0.31 | 0.54 | 0.30 | 0.47 | 0.38 | 0.46 | 0.54 | **0.58** |
| HalfCheetah | 0.48 | 0.49 | 0.47 | 0.49 | 0.50 | 0.49 | 0.50 | **0.50** |
| Average | 0.34 | 0.43 | 0.34 | 0.40 | 0.39 | 0.42 | 0.43 | **0.45** |

### 4.2. Results in Online Continuous Control Tasks

In online continuous control tasks, we explore other dimensions of dynamics shift, namely environment non-stationary and the continuous change of environment parameters. Such tasks are far more complicated than offline tasks with 3 different dynamics, but are within the capability of current approaches thanks to the existence of online interactive training environments. We consider the HalfCheetah, Walker2d, and Ant environments in the MuJoCo simulator (Todorov et al., 2012) and the autonomous driving environment in the MetaDrive simulator (Li et al., 2023). Sources of dynamics change include wind, joint damping, and traffic densities. For baselines we include the IfO algorithm GARAT (Desai et al., 2020), standard online RL algorithm SAC (Haarnoja et al., 2018), online cross-domain policy transfer algorithm OSI (Yu et al., 2017), ESCP (Luo et al., 2022), CaDM (Lee et al., 2020), and SRPO (Xue et al., 2023a).

Comparative results in online continuous control tasks are shown in Fig. 2, where our ESCP+ASOR algorithm has the best performance in all environments. Specifically, it only makes marginal improvements in the HalfCheetah environment, in contrast to large enhancement in other environments. This is because the agent will not "fall over" in the HalfCheetah environment, and the state accessibility will not change a lot under dynamics shift, undermining

the effect of the accessible state-based policy regularization. We also compare in Fig. 3 (left) the augmented reward with the environment reward on different states in Walker-2d. On states where the agent is about to fall over, the augmented reward drops significantly while the environment reward does not change much, demonstrating the effectiveness of the reward augmentation.

### 4.3. Results in A Large-Scale Fall Guys-Like Battle Royal Game Environment

In the large-scale fall guys-like game environment, we focus on highly dynamic and competitive race scenarios, characterized by a myriad of ever-changing obstacles, shifting floor layouts, and functional items. The elements within the game exhibit both functional and attribute changes, resulting in dynamics shift and evolving state accessibility. As shown in Fig. 4, the effects of trampolines (e.g., height and orientation) vary across different maps and the agent's interaction with the trampoline will therefore result in different environment transitions depending on the specific configuration. The resulting dynamics shift has high stochasticity and cannot be effectively modelled by context encoder-based algorithms (Luo et al., 2022; Lee et al., 2020). We train the agent on 10 distinct maps, each presenting unique challenges and configurations. The training step is set to 6M. Metrics except policy entropy were averaged over the final

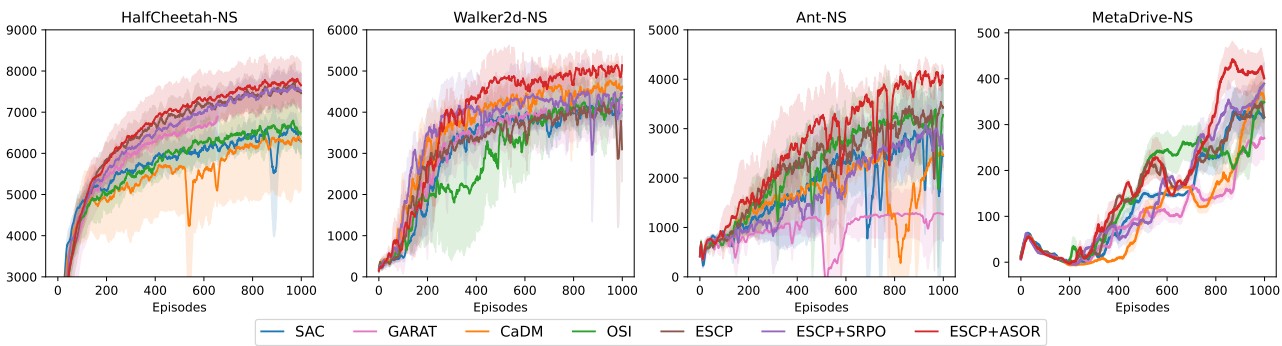

Figure 2. Results of online experiments on MuJoCo and MetaDrive tasks. "NS" refers to tasks with non-stationary environment dynamics.

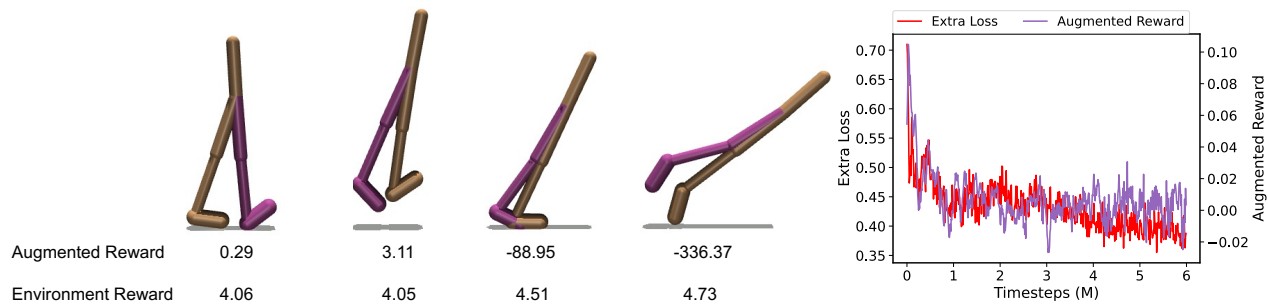

| | | | | |
|---|---|---|---|---|
| Augmented Reward | 0.29 | 3.11 | -88.95 | -336.37 |
| Environment Reward | 4.06 | 4.05 | 4.51 | 4.73 |

Figure 3. **Left**: Comparisons of the logarithm of the discriminator output, i.e., the augmented reward, and the environment reward on different states in the Walker-2d environment. The augmented reward can better reflect the state optimality. **Right**: Curves for average extra loss and augmented reward in the fall-guys like game environment.

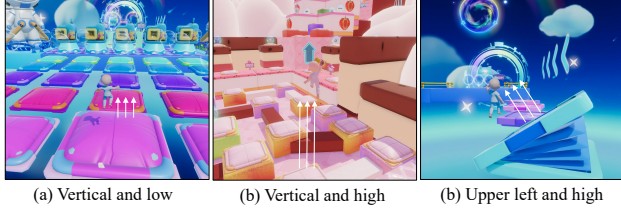

(a) Vertical and low     (b) Vertical and high     (b) Upper left and high

Figure 4. Demonstrations of dynamics shift caused by different trampoline effect. Colors and textures are only for visual enhancement and are not part of the agent's observations.

1M steps and the policy entropy is averaged in the initial 1M steps. More experiment details are listed in Appendix C.3, including additional map demonstrations, MDP setups, and the network structure.

As demonstrated in Tab. 3, PPO+ASOR achieves the highest scores in all five performance-related metrics. To be specific, the high total reward, unweighed goal reward, and success rate demonstrate the overall effectiveness of ASOR when applied to complex large-scale tasks. Low trapped rate, small distance from cliff, and high policy entropy demonstrate the strong exploration ability of ASOR since it is better at getting rid of low-reward regions and has higher policy stochasticity. The low unnecessary jump rate demonstrates the effectiveness of policy regularization only on accessible states. Jumping states may appear in the optimal trajectories

in maps with diverse altitudes, but are unnecessary and hinder the fast goal reaching in other maps. Fig. 3 (right) shows the curve of the augmented reward and the extra loss, including the discriminator training loss and the RND training loss. The loss curve drops smoothly and the average augmented reward remains stable, which means that the discriminator network is easy to train and has stable performance.

## 5. Conclusion

In this paper, we focus on efficient policy optimization with data collected under dynamics shift. We demonstrate that despite widely utilized in IfO algorithms, the idea of similar expert state distribution across different dynamics can be unreliable when some states are no longer accessible as the environment dynamics changes. We propose a policy regularization method that only imitates expert state distributions on globally accessible states. By formally characterizing the difference of state accessibility under dynamics shift, we show that the accessible state-based regularization approach provides strong lower-bound performance guarantees for efficient policy optimization. We also propose a practical algorithm called ASOR that can serve as an add-on reward augmentation module to existing RL approaches. Extensive experiments across various online and offline RL benchmarks indicate ASOR can be effectively integrated with several state-of-the-art cross-domain policy transfer algorithms, substantially enhancing their performance.

*Table 3.* Experiment results in the fall guys-like game environment. Metrics with the up arrow ($\uparrow$) are expected to have larger values and vice versa. Metrics with ($\sim$) have no specific tendencies.

| | Total Reward ($\uparrow$) | Goal Reward ($\uparrow$) | Success Rate ($\uparrow$) | Trapped Rate ($\downarrow$) | Unnecessary Jump Rate ($\downarrow$) | Distance from Cliff ($\sim$) | Policy Entropy ($\sim$) |
|---|---|---|---|---|---|---|---|
| PPO | 0.329±0.308 | 1.154±0.085 | 0.361±0.009 | 0.012±0.009 | 0.064±0.003 | 0.152±0.025 | 5.725±0.185 |
| PPO+SRPO | 0.337±0.257 | 1.513±0.076 | 0.376±0.006 | 0.038±0.015 | 0.040±0.003 | 0.148±0.018 | 5.859±0.098 |
| PPO+ASOR | **0.554**±0.336 | **1.781**±0.053 | **0.387**±0.005 | **0.005**±0.005 | **0.029**±0.003 | 0.143±0.021 | 6.358±0.122 |

**Limitations** This paper focuses on the setting of HiP-MDP with evolving environment dynamics and a static reward function. The resulting ASOR algorithm will not be applicable to tasks with multiple reward functions. Meanwhile, the theoretical results will be weaker on some adversarial HiP-MDPs with large $R_s$. Details will be discussed in Appendix A.3.

## Impact Statement

The ASOR approach enables better adaptation of RL models in dynamic environments, making RL applications more resilient across diverse real-world conditions. The concept of avoiding misleading inaccessible states may improve the safety of autonomous agents, ensuring that they do not make catastrophic decisions when transitioning between environments. Therefore, ASOR's method can enhance cross-domain policy transfer, making RL algorithms more efficient for applications such as robotics, industrial automation, healthcare, and finance.

## Acknowledgement

This research is supported by the National Research Foundation, Singapore under its Industry Alignment Fund – Prepositioning (IAF-PP) Funding Initiative. Any opinions, findings and conclusions or recommendations expressed in this material are those of the author(s) and do not reflect the views of National Research Foundation, Singapore. This research is also supported by the Joint NTU-WeBank Research Centre on Fintech, Nanyang Technological University, Singapore.

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

## A. Additional Derivations and Proofs

### A.1. Derivations of the Lagrangian

For expression convenience, we denote $d_T^\pi(\cdot)$ with $\mu(\cdot)$ and $d_{T_0}^{*,+}(\cdot)$ with $\nu(\cdot)$. We also omit the maximization over $T$ in Eq. (1) as it can be obtained by following all policy constrains in different dynamics. We start from the optimization problem

$$\max_\pi \ \mathbb{E}_{s_t,a_t,s_{t+1}\sim\tau_\pi}\left[\sum_{t=0}^\infty \gamma^t r\left(s_t,a_t,s_{t+1}\right)\right] \quad \text{s.t.} \ d_{\mathcal{F},\phi}\left(\mu(\cdot),\nu(\cdot)\right) < \varepsilon. \tag{6}$$

The $\mathcal{F}$-distance term can be transformed as:

$$\begin{aligned}
d_{\mathcal{F},\phi}\left(\mu(\cdot),\nu(\cdot)\right) &= \mathbb{E}_{s\sim\mu}[\log\omega^*(s)] + \mathbb{E}_{s\sim\nu}[\log(1-\omega^*(s))] \\
&= \int \mu(s)\log\omega^*(s)ds + \int \nu(s)\log(1-\omega^*(s))ds \\
&= \int (1-\gamma)\sum_{t=0}^\infty \gamma^t p(s_t=s|\pi,T)\log\omega^*(s)ds \\
&\quad + \int (1-\gamma)\sum_{t=0}^\infty \gamma^t p(s_t=s, s_t\in S^+|\pi^*,T_0)\log(1-\omega^*(s))ds \\
&= (1-\gamma)\sum_{t=0}^\infty \gamma^t \left[\mathbb{E}_{s_t\sim\tau_\pi}\log\omega^*(s_t) + \mathbb{E}_{s_t\sim\tau^{*,+}}\log(1-\omega^*(s_t))\right]
\end{aligned} \tag{7}$$

where $\omega^*$ is the trained discriminator. So the optimization problem can be written as the following standard form

$$\begin{aligned}
\min_\pi \ & \mathbb{E}_{s_t,a_t,s_{t+1}\sim\tau_\pi}\sum_{t=0}^\infty -\gamma^t r\left(s_t,a_t,s_{t+1}\right) \\
\text{s.t.} \ & -(1-\gamma)\sum_{t=0}^\infty \gamma^t\left[\mathbb{E}_{s_t\sim\tau_\pi}\log\omega^*(s_t) + \mathbb{E}_{s_t\sim\tau^{*,+}}\log(1-\omega^*(s_t))\right] - \frac{\varepsilon}{1-\gamma} < 0.
\end{aligned} \tag{8}$$

Neglecting items irrelevant to $\pi$, we get the Lagrangian $L$ as

$$L = -\mathbb{E}_{s_t,a_t,s_{t+1}\sim\tau_\pi}\left[\sum_{t=0}^\infty \gamma^t\left(r(s_t,a_t,s_{t+1}) + \lambda\log\omega^*(s_t)\right)\right] - \frac{\lambda\varepsilon}{1-\gamma}. \tag{9}$$

### A.2. Proofs of Theorems

**Lemma A.1** (Value Discrepancy). *Considering MDPs $\mathcal{M}_1=(\mathcal{S},\mathcal{A},T_1,r,\gamma,\rho_0)$ and $\mathcal{M}_2=(\mathcal{S},\mathcal{A},T_2,r,\gamma,\rho_0)$ which are $M$-$R_s$ accessible from each other, for all $s\in\mathcal{S}$ we have*

$$|V_{T_1}^*(s) - V_{T_2}^*(s)| \leqslant \frac{R_s+2\lambda}{1-\gamma}, \tag{10}$$

*where $\lambda$ is the action coefficient in the reward function. Detailed definition are in Sec. 2.1.*

*Proof.* Without the loss of generality, we consider the state $s$ with $V_{T_1}^*(s) \geqslant V_{T_2}^*(s)$. Under the optimal policy $\pi_1^*(s)$, the next state of $s$ in $\mathcal{M}_1$ will be $s' = T(s,a^*)$. As $\mathcal{M}_2$ is $M$-$R_s$ accessible from $\mathcal{M}_1$, there exists $N\leqslant M$ such that in $\mathcal{M}_2$, $s'$ can be reached from $s$ with action sequence $a_1,a_2,\cdots,a_N$. We then borrow the idea of iteratively computing $|V_{T_1}^*(s) - V_{T_2}^*(s)|$ from Xue et al. (Xue et al., 2023a). According to the optimistic Bellman equation

$$V_T^*(s) = \max_a \ r(s,a,s') + \gamma V_T^*(T(s,a)), \tag{11}$$

we have

$$
\begin{aligned}
&\left|V_{T_1}^*(s) - V_{T_2}^*(s)\right| \\
&= V_{T_1}^*(s) - V_{T_2}^*(s) \\
&\leqslant r(s, a^*, s') + \gamma V_{T_1}^*(s') - \sum_{i=0}^{N-1} \gamma^i r(s_i, a_i, s_{i+1}) - \gamma^N V_{T_2}^*(s') \\
&\quad (s_0 \doteq s, \ s_N \doteq s' \text{ for brevity}) \\
&= r(s, a^*, s') - \gamma^{N-1} r(s_{N-1}, a_{N-1}, s') + \gamma(1 - \gamma^{N-1}) V_{T_2}^*(s') \\
&\quad + \sum_{n=0}^{N-2} \gamma^n r(s_n, a_n, s_{n+1}) + \gamma[V_{T_1}^*(s') - V_{T_2}^*(s')] \\
&\leqslant (1 - \gamma^{N-1})(r(s, a^*, s') + \gamma V_{T_2}^*(s')) + 2\lambda + \sum_{n=0}^{N-2} \gamma^n r(s_n, a_n, s_{n+1}) + \gamma[V_{T_1}^*(s') - V_{T_2}^*(s')] \\
&\leqslant (1 - \gamma^{N-1})(r(s, a^*, s') + \gamma V_{T_1}^*(s')) + 2\lambda + \sum_{n=0}^{N-2} \gamma^n r(s_n, a_n, s_{n+1}) + \gamma[V_{T_1}^*(s') - V_{T_2}^*(s')] \\
&\leqslant R_s + 2\lambda + \gamma[V_{T_1}^*(s') - V_{T_2}^*(s')].
\end{aligned}
\tag{12}
$$

Iteratively computing $\left|V_{T_1}^*(s) - V_{T_2}^*(s)\right|$, we have

$$
\left|V_{T_1}^*(s') - V_{T_2}^*(s')\right| \leqslant \frac{R_s + 2\lambda}{1 - \gamma}.
\tag{13}
$$

$\square$

**Theorem A.2** (Thm. 3.5 in the main paper.). *Consider the MDP $\mathcal{M}_1 = (\mathcal{S}, \mathcal{A}, T_1, r, \gamma, \rho_0)$ which is $M$-$R_s$ accessible from the MDP $\mathcal{M}_2 = (\mathcal{S}, \mathcal{A}, T_2, r, \gamma, \rho_0)$. For all policy $\hat{\pi}$, if there exists one certain dynamics $T_0$ such that $\max_{T} D_{\mathrm{JS}}(d_T^{\hat{\pi}}(\cdot) \| d_{T_0}^{*;+}(\cdot)) \leqslant \varepsilon$, we have*

$$
\eta(\hat{\pi}) \geqslant \max_{T} \eta(\pi_T^*) - \frac{2R_s + 6\lambda + 2R_{\max}\varepsilon}{1 - \gamma}.
\tag{14}
$$

*Proof.* $\left|\eta_{T_1}(\pi_{T_1}^*) - \eta_{T_2}(\pi_{T_2}^*)\right|$ can be bounded with Thm. A.1:

$$
\left|\eta_{T_1}(\pi_{T_1}^*) - \eta_{T_2}(\pi_{T_2}^*)\right| = \left|\mathbb{E}_{s \in \rho_0} V_{T_1}^*(s) - \mathbb{E}_{s \in \rho_0} V_{T_2}^*(s)\right| \leqslant \frac{R_s + 2\lambda}{1 - \gamma}.
\tag{15}
$$

With a slight abuse of notation, we define the transition distribution $d_T^\pi(s, a, s') = d_T^\pi(s)\pi(a|s)T(s'|s, a)$ and the accessible-state transition distribution $d_{T_2}^{\pi,+}(s, a, s') = d_T^{\pi,+}(s)\hat{\pi}(a|s)T(s'|s, a)$. Consider $\tilde{\pi}$ such that $d_T^{\tilde{\pi}}(s) = d_{T_0}^{*;+}(s)$ for all $s \in \mathcal{S}$. The accumulated return of policy $\tilde{\pi}$ under transition $T_1$ can be written as $\eta_{T_1}(\hat{\pi}) = (1 - \gamma)^{-1}\mathbb{E}_{s,a,s' \sim d_{T_1}^{\tilde{\pi}}}[r(s, a, s')]$. We also consider the accumulated return of the optimal policy under transition $T_2$ including only accessible states: $\eta_{T_2}^+(\pi_{T_2}^{*,+}) = (1 - \gamma)^{-1}\mathbb{E}_{s,a,s' \sim d_{T_2}^{*,+}}[r(s, a, s')]$, where $\pi_{T_2}^{*,+}$ is the optimal policy making transitions among accessible states. Consider the Lipschitz property of the reward function:

$$
|r(s, a_1, s') - r(s, a_2, s')| \leqslant \lambda \|a_1 - a_2\|_1.
\tag{16}
$$

Taking expectation w.r.t. $d_{T_1}^{\tilde{\pi}}(\cdot)$ on both sides, we get

$$
\mathbb{E}_{s \sim d_{T_1}^{\tilde{\pi}}} |r(s, a_1, s') - r(s, a_2, s')| \leqslant \mathbb{E}_{s \sim d_{T_1}^{\tilde{\pi}}} \lambda \|a_1 - a_2\|_1.
\tag{17}
$$

Letting $\mu(A_1, A_2|s)$ be any joint distribution with marginals $\hat{\pi}$ and $\pi_{T_2}^{*,+}$ conditioned on $s \in S^+$. Taking expectation w.r.t. $\mu$

on both sides, we get

$$
\begin{aligned}
\left| \mathbb{E}_{d_{T_1}^{\hat{\pi}}} r(s,a,s') - \mathbb{E}_{d_{T_2}^{*,+}} r(s,a,s') \right| &\leqslant \mathbb{E}_{s \sim d_{T'}^*} \mathbb{E}_{a_1,a_2 \sim \mu} |r(s,a_1,s') - r(s,a_2,s')| \\
&\leqslant \lambda \mathbb{E}_{s \sim d_{T_1}^{\hat{\pi}}} E_\mu \|a_1 - a_2\|_1 \\
&\leqslant \max_s \lambda E_\mu \|a_1 - a_2\|_1 \\
&\leqslant 2\lambda
\end{aligned}
\tag{18}
$$

According to the definitions of $\eta_{T_1}(\tilde{\pi})$ and $\eta_{T_2}^+(\pi_{T_2}^{+,*})$, the L.H.S. of Eq. (18) is exactly the difference of the two accumulated returns. Therefore, we get

$$
\left| \eta_{T_1}(\tilde{\pi}) - \eta_{T_2}^+(\pi_{T_2}^{*,+}) \right| \leqslant \frac{2\lambda}{1-\gamma}.
\tag{19}
$$

Then we will compute the discrepancy between $\eta_{T_2}^+$ and $\eta_{T_2}$. $\eta_{T_2}$ can be computed with

$$
\begin{aligned}
\eta_{T_2}(\pi_{T_2}^*) &= \mathbb{E}_{s \sim \rho_0} V_{T_2}^{\pi_{T_2}^*}(s) \\
&= \mathbb{E}_\tau \sum_{n=0}^{N-1} \gamma^n r(s_n,a_n,s_{n+1}) + \gamma^N V_{T_2}^{\pi_{T_2}^*}(s_N),
\end{aligned}
\tag{20}
$$

where $s_N$ is the accessible state accessible from $s_0$ with $\pi_{T_2}^{*,+}$. According to the definition of $M$-$R_s$ accessible MDPs,

$$
\begin{aligned}
\eta_{T_2}(\pi_{T_2}^*) &= \mathbb{E}_\tau \sum_{n=0}^{N-1} \gamma^n r_n + \gamma^N V_{T_2}^{\pi_{T_2}^*}(s_N) - R_s + R_s \\
&\leqslant \mathbb{E}_\tau \sum_{n=0}^{N-1} \gamma^n r_n - \sum_{n=0}^{N-2} \gamma^n r_n - (\gamma^{N-1}-1) r(s_0, \pi_{T_2}^{+,*}(s_0), s_N) \\
&\quad + \gamma^N V_{T_2}^{\pi_{T_2}^*}(s_N) - (\gamma^N - \gamma) V_{T_2}^{\pi_{T_2}^{+,*}}(s_N) + R_s \\
&\leqslant \mathbb{E}_\tau r(s_0, \pi_{T_2}^{+,*}(s_0), s_N) + \gamma V_{T_2}^{\pi_{T_2}^{+,*}}(s_N) + \gamma^N (V_{T_2}^{\pi_{T_2}^*}(s_N) - V_{T_2}^{\pi_{T_2}^{+,*}}(s_N)) + R_s + 2\lambda \\
&= \eta_{T_2}^+(\pi_{T_2}^{+,*}) + \gamma^N (V_{T_2}^{\pi_{T_2}^*}(s_N) - V_{T_2}^{\pi_{T_2}^{+,*}}(s_N)) + R_s + 2\lambda,
\end{aligned}
\tag{21}
$$

where $r_n$ is the short for $r(s_n,a_n,s_{n+1})$. Iteratively scaling the value discrepancy between $\pi_{T_2}^*$ and $\pi_{T_2}^{+,*}$, we get

$$
\left| \eta_{T_2}(\pi_{T_2}^*) - \eta_{T_2}^+(\pi_{T_2}^{+,*}) \right| \leqslant \frac{R_s + 2\lambda}{1-\gamma^M} \leqslant \frac{R_s + 2\lambda}{1-\gamma}.
\tag{22}
$$

According to results in imitation learning (Lem. 6 in (Xu et al., 2020)), we have

$$
\left| \eta_{T_1}(\tilde{\pi}) - \eta_{T_1}(\hat{\pi}) \right| \leqslant \frac{2R_{\max}\varepsilon}{1-\gamma}
\tag{23}
$$

Combining Eq. (15)(19)(21)(23), we have

$$
\begin{aligned}
\left| \eta_{T_1}(\tilde{\pi}) - \eta_{T_1}(\pi_{T_1}^*) \right| &\leqslant \left| \eta_{T_1}(\hat{\pi}) - \eta_{T_1}(\tilde{\pi}) \right| + \left| \eta_{T_1}(\tilde{\pi}) - \eta_{T_2}^+(\pi_{T_2}^{+,*}) \right| \\
&\quad + \left| \eta_{T_2}^+(\pi_{T_2}^{+,*}) - \eta_{T_2}(\pi_{T_2}^*) \right| + \left| \eta_{T_2}(\pi_{T_2}^*) - \eta_{T_1}(\pi_{T_1}^*) \right| \\
&\leqslant \frac{2R_s + 6\lambda + 2R_{\max}\varepsilon}{1-\gamma}.
\end{aligned}
\tag{24}
$$

Taking expectation with respect to all $T$ in the HiP-MDP concludes the proof. $\qquad\square$

**Lemma A.3** (Lemma 2 in Xu et. al (Xu et al., 2020)). *Consider a network class set $\mathcal{P}$ with $\Delta$-bounded value functions, i.e., $|P(s)| \leq \Delta$, for all $s \in \mathcal{S}, P \in \mathcal{P}$. Given an expert policy $\pi_E$ and an imitated policy $\pi_I$ with $d_\mathcal{P}\left(\hat{d}^{\pi_E}, \hat{d}^{\pi_I}\right) - \inf_{\pi \in \Pi} d_\mathcal{P}\left(\hat{d}^{\pi_E}, \hat{d}^\pi\right) \leq \varepsilon_\mathcal{P}$, then $\forall \delta \in (0,1)$, with probability at least $1-\delta$, we have*

$$
d_\mathcal{P}\left(d^{\pi_E}, d^{\pi_I}\right) \leq \inf_{\pi \in \Pi} d_\mathcal{P}\left(\hat{d}^{\pi_E}, \hat{d}^\pi\right) + 2\hat{\mathcal{R}}_{d^{\pi_E}}^{(m)}(\mathcal{P}) + 2\hat{\mathcal{R}}_{d^{\pi_I}}^{(m)}(\mathcal{P}) + 12\Delta\sqrt{\frac{\log(2/\delta)}{m}} + \varepsilon_\mathcal{P}.
\tag{25}
$$

*Proof.* See Appendix B.3 of (Xu et al., 2020). □

**Theorem A.4.** *Consider the MDP $\mathcal{M}_1 = (\mathcal{S}, \mathcal{A}, T_1, r, \gamma, \rho_0)$ which is $M$-$R_s$ accessible from the MDP $\mathcal{M}_2 = (\mathcal{S}, \mathcal{A}, T_2, r, \gamma, \rho_0)$ and the network set $\mathcal{P}$ bounded by $\Delta$, i.e., $|P(s)| \leqslant \Delta$. Given $\{s^{(i)}\}_{i=1}^m$ sampled from $d_{T_2}^{+,*}$, if $\pi_{T_2}^{+,*} \in \mathcal{P}$ and the reward function $r_{\hat{\pi}, T_1}(s) = \mathbb{E}_{a \sim \hat{\pi}, s' \sim T_1} r(s, a, s')$ lies in the linear span of $\mathcal{P}$, for policy $\hat{\pi}$ regularized by $\hat{d}_{T_2}^{+,*}$ according to Eq. (1) with $d_\mathcal{P}(\hat{d}_{T_1}^{\hat{\pi}}, \hat{d}_{T_2}^{+,*}) < \varepsilon_\mathcal{P}$, we have*

$$\eta_{T_1}(\hat{\pi}) \geqslant \eta_{T_1}(\pi_{T_1}^*) - \frac{2R_s + 8\lambda}{1 - \gamma} - \frac{2\|r\|_\mathcal{P}}{1 - \gamma} \left( \hat{\mathcal{R}}_{d_{T_2}^{+,*}}^{(m)}(\mathcal{P}) + \hat{\mathcal{R}}_{d_{T_1}^{\hat{\pi}}}^{(m)}(\mathcal{P}) + 6\Delta \sqrt{\frac{\log(2/\delta)}{m}} + \frac{\varepsilon}{2} \right) \tag{26}$$

*with probability at least $1 - \delta$.*

*Proof.* As $\mathcal{M}_1$ is $M$-$R_s$ accessible accessible from $\mathcal{M}_2$, there exists policy $\tilde{\pi}$ such that $d_{T_1}^{\tilde{\pi}}(s) = d_{T_2}^{*,+}(s)$ for all $s \in \mathcal{S}^+$. With Thm. A.2, we have

$$\eta_{T_1}(\tilde{\pi}) \geqslant \eta_{T_1}(\pi_{T_1}^*) - \frac{2R_s + 6\lambda}{1 - \gamma} \tag{27}$$

Then we compute the performance discrepancy $\eta_{T_1}(\hat{\pi}) - \eta_{T_1}(\tilde{\pi})$ given that $d_\mathcal{P}(\hat{d}_{T_1}^{\hat{\pi}}, \hat{d}_{T_1}^{\tilde{\pi}}) < \varepsilon_\mathcal{P}$. The following derivations borrow the main idea from Xu et al. (Xu et al., 2020) and turn the state-action occupancy measure $\rho$ into the state-only occupancy measure $d$. We start with the network distance of the ground truth state occupancy measures. According to Lem. A.3, we have

$$d_\mathcal{P}(d_{T_1}^{\hat{\pi}}, d_{T_1}^{\tilde{\pi}}) \leqslant 2\hat{\mathcal{R}}_{d_{T_2}^{+,*}}^{(m)}(\mathcal{P}) + 2\hat{\mathcal{R}}_{d_{T_1}^{\hat{\pi}}}^{(m)}(\mathcal{P}) + 12\Delta \sqrt{\frac{\log(2/\delta)}{m}} + \varepsilon_\mathcal{P} \tag{28}$$

with probability at least $1 - \delta$. Meanwhile,

$$
\begin{aligned}
&|\eta_{T_1}(\hat{\pi}) - \eta_{T_1}(\tilde{\pi})| \\
&\leqslant \frac{1}{1 - \gamma} \left| \mathbb{E}_{s \sim d_{T_1}^{\hat{\pi}}} [r_{\hat{\pi}, T_1}(s)] - \mathbb{E}_{s \sim d_{T_1}^{\tilde{\pi}}} [r_{\tilde{\pi}, T_1}(s)] \right| \\
&\leqslant \frac{1}{1 - \gamma} \left| \mathbb{E}_{s \sim d_{T_1}^{\hat{\pi}}} [r_{\hat{\pi}, T_1}(s)] - \mathbb{E}_{s \sim d_{T_1}^{\tilde{\pi}}} [r_{\hat{\pi}, T_1}(s)] \right| + \frac{1}{1 - \gamma} \left| \mathbb{E}_{s \sim d_{T_1}^{\tilde{\pi}}} [r_{\hat{\pi}, T_1}(s)] - \mathbb{E}_{s \sim d_{T_1}^{\tilde{\pi}}} [r_{\tilde{\pi}, T_1}(s)] \right| \\
&\leqslant \frac{1}{1 - \gamma} \left| \mathbb{E}_{s \sim d_{T_1}^{\hat{\pi}}} [r_{\hat{\pi}, T_1}(s)] - \mathbb{E}_{s \sim d_{T_1}^{\tilde{\pi}}} [r_{\hat{\pi}, T_1}(s)] \right| + \frac{2\lambda}{1 - \gamma}.
\end{aligned}
\tag{29}
$$

As we assume that the reward function $r_{\hat{\pi}, T_1}(s)$ lies in the linear span of $\mathcal{P}$, there exists $n \in \mathbb{N}, \{c_i \in \mathbb{R}\}_{i=1}^n$ and $\{P_i \in \mathcal{P}\}_{i=1}^n$, such that $r = c_0 + \sum_{i=1}^n c_i P_i$. So we obtain that

$$
\begin{aligned}
|\eta_{T_1}(\hat{\pi}) - \eta_{T_1}(\tilde{\pi})| &\leqslant \frac{1}{1 - \gamma} \left| \mathbb{E}_{s \sim d_{T_1}^{\hat{\pi}}} [r_{\hat{\pi}, T_1}(s)] - \mathbb{E}_{s \sim d_{T_1}^{\tilde{\pi}}} [r_{\hat{\pi}, T_1}(s)] \right| + \frac{2\lambda}{1 - \gamma} \\
&\leqslant \frac{1}{1 - \gamma} \left| \sum_{i=1}^n c_i \mathbb{E}_{s \sim d_{T_1}^{\hat{\pi}}} [P_i(s, a)] - \sum_{i=1}^n c_i \mathbb{E}_{s \sim d_{T_1}^{\tilde{\pi}}} [P_i(s, a)] \right| + \frac{2\lambda}{1 - \gamma} \\
&\leqslant \frac{1}{1 - \gamma} \sum_{i=1}^n |c_i| \left| \mathbb{E}_{s \sim d_{T_1}^{\hat{\pi}}} [P_i(s, a)] - \mathbb{E}_{s \sim d_{T_1}^{\tilde{\pi}}} [P_i(s, a)] \right| + \frac{2\lambda}{1 - \gamma} \\
&\leqslant \frac{1}{1 - \gamma} \left( \sum_{i=1}^n |c_i| \right) d_\mathcal{P} \left( d_{T_1}^{\hat{\pi}}, d_{T_1}^{\tilde{\pi}} \right) + \frac{2\lambda}{1 - \gamma} \\
&\leqslant \frac{1}{1 - \gamma} \|r\|_\mathcal{P} d_\mathcal{P} \left( d_{T_1}^{\hat{\pi}}, d_{T_1}^{\tilde{\pi}} \right) + \frac{2\lambda}{1 - \gamma}.
\end{aligned}
\tag{30}
$$

Combining Eq. (28) and Eq. (30), we have

$$\eta_{T_1}(\hat{\pi}) \geqslant \eta_{T_1}(\tilde{\pi}) - \frac{2\|r\|_\mathcal{P}}{1 - \gamma} \left( \hat{\mathcal{R}}_{d_{T_2}^{+,*}}^{(m)}(\mathcal{P}) + \hat{\mathcal{R}}_{d_{T_1}^{\hat{\pi}}}^{(m)}(\mathcal{P}) + 6\Delta \sqrt{\frac{\log(2/\delta)}{m}} + \frac{\varepsilon}{2} \right) + \frac{2\lambda}{1 - \gamma} \tag{31}$$

*Table 4.* Comparison between the Lipschitz coefficient $\lambda$ and the maximum reward $R_{\max}$ in practical environments.

| Environment | Action-related Reward | $\lambda$ | $R_{\max}$ |
|---|---|---|---|
| CartPole-v0 | 0 | 0 | 1.00 |
| InvertedPendulum-v2 | 0 | 0 | 1.00 |
| Lava World | 0 | 0 | 1.00 |
| MetaDrive | 0 | 0 | $\geqslant 1$ |
| Fall-guys Like Game | 0 | 0 | $\geqslant 1$ |
| Swimmer-v2 | $-0.0001\|a\|_2^2$ | 0.0001 | 0.36 |
| HalfCheetah-v2 | $-0.1\|a\|_2^2$ | 0.1 | 4.80 |
| Hopper-v2 | $-0.001\|a\|_2^2$ | 0.001 | 3.80 |
| Walker2d-v2 | $-0.001\|a\|_2^2$ | 0.001 | $\geqslant 4$ |
| Ant-v2 | $-0.5\|a\|_2^2$ | 0.5 | 6.00 |
| Humanoid-v2 | $-0.1\|a\|_2^2$ | 0.1 | $\geqslant 8$ |

with probability at least $1 - \delta$. Combining Eq. (31) and Eq. (27), we have

$$\eta_{T_1}(\hat{\pi}) \geqslant \eta_{T_1}(\pi_{T_1}^*) - \frac{2R_s + 8\lambda}{1 - \gamma} - \frac{2\|r\|_{\mathcal{P}}}{1 - \gamma}\left(\hat{\mathcal{R}}_{d_{T_2}^{+,*}}^{(m)}(\mathcal{P}) + \hat{\mathcal{R}}_{d_{T_1}^{\hat{\pi}}}^{(m)}(\mathcal{P}) + 6\Delta\sqrt{\frac{\log(2/\delta)}{m}} + \frac{\varepsilon}{2}\right) \tag{32}$$

with probability at least $1 - \delta$. Taking expectation with respect to all $T$ in the HiP-MDP concludes the proof. $\qquad\square$

### A.3. Discussions on the Theorems

**Lipschitz Assumption** The Lipschitz assumption in Sec. 2.1 requires that if $s$ and $s'$ keep unchanged, the deviation of the reward $r$ will not be larger than $\lambda$ times the deviation of the action $a$:

$$|r(s, a_1, s') - r(s, a_2, s')| \leqslant \lambda\|a_1 - a_2\|_1. \tag{33}$$

Therefore, the Lipschitz coefficient $\lambda$ is only depends action-related terms in the reward function. In Tab. 4, we list the action-related terms of the reward functions for various RL evaluation environments, along with the corresponding values of $\lambda$ derived from these terms. As indicated in the table, the action-related terms in reward functions exhibit reasonably small coefficients in all environments compared with the maximum environment reward $R_{\max}$. Therefore, the Lipschitz coefficient $\lambda$ will not dominate the error term in Thm. 3.5 and Thm. 3.7.

**Failure Cases** Apart from the Lipschitz assumptions that can easily be realized, Thm. 3.5 and Thm. 3.7 depend on the formulation of $M$-$R_s$ accessible MDPs. Potential failure cases will therefore include tasks with high $R_s$, so that the performance lower bounds become weak. This will happen if states with lowest rewards exist in the optimal trajectory of some, and not all dynamics. In the example of lava world in Fig. 1, if a reward of -100 is assigned to grid (3,4), $R_s$ will be as large as 100, leading to a weak performance lower bound when the bottom lava block is at Row 2 with a best episode return of 1. Nevertheless, issues with theoretical analyses will not negatively influence the practical performance of ASOR.

### A.4. Comparisons with Previous Approaches

Intuitively, the practical algorithm procedure of ASOR share some insights with some offline RL algorithms including AWAC (Nachum et al., 2019), CQL (Kumar et al., 2020) and MOPO (Yu et al., 2020). For example, ASOR prefers states with high values similar to AWAC and states with high visitation counts similar to CQL and MOPO. The advances of ASOR include: 1) By restricting the considered states to accessible states, ASOR can be applied in offline datasets collected under dynamics shift, where the aforementioned offline RL algorithms can only learn from the dataset with static dynamics. Thm. 3.5 and Thm. 3.7 demonstrate the effectiveness of such procedure. 2) ASOR modifies the original policy optimization process by reward augmentation. This enables the easy combination of ASOR with other cross-domain algorithms to enhance their performance.

ASOR also share the approach of classifier-based reward augmentation with DARA (Liu et al., 2022) and SRPO (Xue et al., 2023a). The classifier input in DARA is $(s, a, s')$ from the source and target environments. Compared with ASOR, the

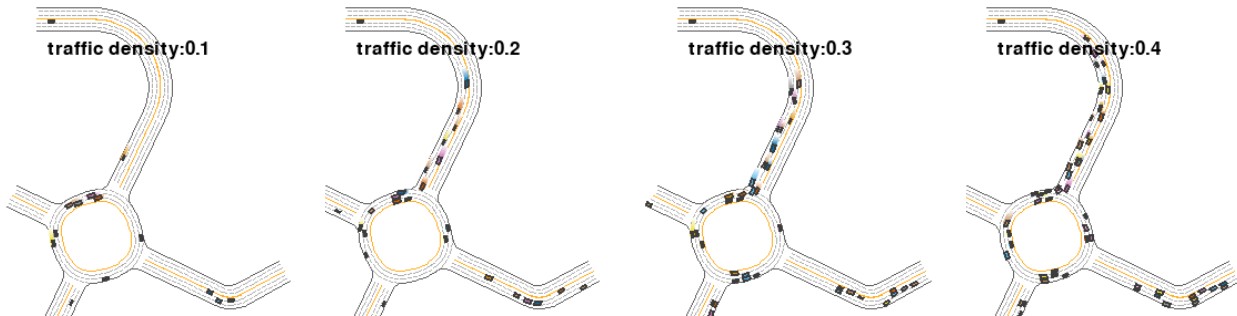

*Figure 5.* MetaDrive environments with different traffic densities.

classifier in DARA exhibits higher complexity and is harder to train. It also requires the access to the information of target environments. Therefore, DARA has poor performance as demonstrated in Sec. 4.1 and cannot be applied to tasks with no prior knowledge on the target environments. The algorithm and theories of SRPO are based on the assumption of the same state accessibility, which is an over-simplification of some environments, as demonstrated in Sec. 3.1 and Sec. B. Comparative results in Sec. 4 demonstrate the inferior performance of SRPO compared with ASOR, in correspondence with the flaw in the assumption. Also, the theoretical analysis in the SRPO paper is built on the assumption called "homomorphous MDPs" which is stronger than the $M$-$R_s$ accessible MDPs used in this paper and is a special case of the latter.

## B. Examples of Distinct State Distributions

We claim in the main paper that previous assumption of similar state distributions under distribution shift will not hold in many scenarios. Apart from the motivating example of lava world in Sec. 3.1, we demonstrate more examples in the MetaDrive (Li et al., 2023) and the fall-guys like game environment. Examples of MetaDrive environments with different traffic densities are shown in Fig. 5. The dynamics shift lies in that the ego vehicle will have different probabilities to detect other vehicles nearby. In environments with low traffic densities, there is enough space for some vehicles with optimal policies to drive in high speeds. But in environments packed with surrounding vehicles, fast driving will surely lead to collisions. So the vehicles can only drive in low speeds. As the vehicle speed is included in the agent's state space, difference in traffic densities will lead to distinct optimal state distributions.

Visualizations of the fall-guys like game environment used in Sec. 4.3 are shown in Fig. 7, where map components including the conveyor belt speed, the balloon reaction, the floor reaction, and the hammer distance will work together, giving rise to dynamics shift. Taking the variation of hammer distance (Fig. 7 (d)) as an example, in the left environment the optimal trajectory will contain states where the hammer is near the agent. But in the right environment, there are trajectories that keep the hammer far away to avoid being hit out of the playground. Blindly imitating optimal states collected in the left environment will lead to suboptimal performance in the right environment.

## C. Experiment Details

### C.1. Baseline Algorithms

In experiments with four different tasks, we compare ASOR with the following baseline algorithms:

- PPO (Schulman et al., 2017): The widely used, off-the-shelf online RL algorithm with on-policy policy update.

- SAC (Haarnoja et al., 2018): The widely used off-policy RL algorithm with entropy maximization for better exploration.

- BCO (Torabi et al., 2018a): Learn a agent-specific inverse dynamics model to infer the experts' missing action information.

- GAIfO (Torabi et al., 2018b): A state-only version of the GAIL algorithm.

- GARAT (Desai et al., 2020): Use the action transformer trained with GAIL-like imitation learning to recover the experts' next states in the original environment.

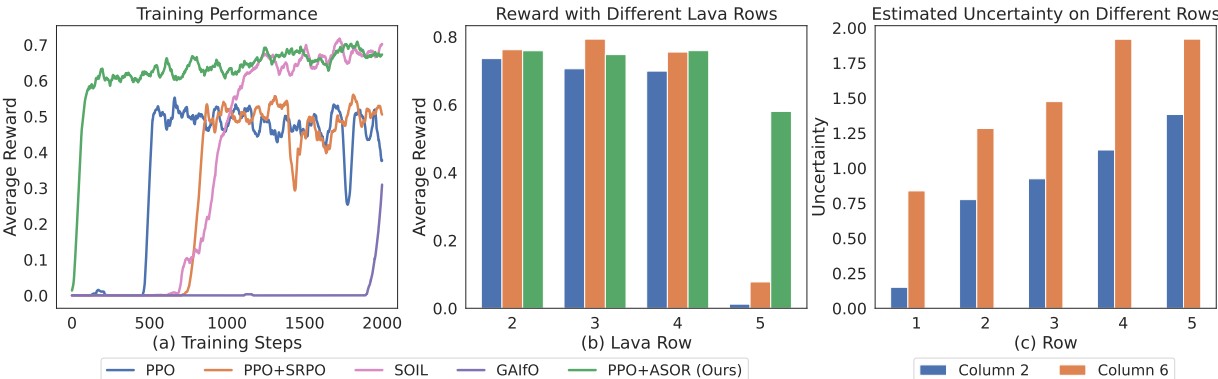

*Figure 6.* Results in the Minigrid environment. (a) Performance comparison between PPO+ASOR and baseline algorithms. (b) The average reward on the environment with different position of the second lava row. PPO and PPO+SRPO has very low rewards when the bottom lava is on row 5; (c) The state uncertainty estimated by ASOR on different rows of the lava environment.

- SOIL (Gangwani & Peng, 2020): An algorithm combining state-only imitation learning with policy gradients. The overall gradient consists of a policy gradient term and an auxiliary imitation term.

- CQL (Kumar et al., 2020): The widely used offline RL algorithm with conservative Q-learning.

- MOPO (Yu et al., 2020): A model-based offline RL algorithm subtracting disagreements in next-state prediction from environment rewards.

- MAPLE (Chen et al., 2021): The offline RL algorithm based on MOPO with an additional context encoder module for cross-dynamics policy adaptation.

- OSI (Yu et al., 2017): An algorithm using context encoders for online system identification.

- CaDM (Lee et al., 2020): The online RL algorithm with context encoders for cross-dynamics policy adaptation.

- DARA (Liu et al., 2022): Make reward augmentations with importance weights between source and target dynamics.

- SRPO (Xue et al., 2023a): Make reward augmentations with the assumption of similar optimal state distributions under dynamics shift.

### C.2. Results in Minigrid Environment

For experiments in the Minigrid environment (Chevalier-Boisvert et al., 2023), the row number of the bottom lava is randomly sampled from $\{2, 3, 4, 5\}$, leading to dynamics shift. By including lava indicator as part of the state input, the policy is fully aware of environment dynamics changes and the need of context encoders (Luo et al., 2022; Lee et al., 2020) is excluded. The categorical action space includes moving towards four directions. The reward function for each environment step is -0.02 and reaching the green goal grid will lead to an additional reward of 1. The episode terminates when the red lava or the green goal grid is reached. For baseline algorithms we select online RL algorithms PPO (Schulman et al., 2017) and PPO+SRPO (Xue et al., 2023a), as well as IfO algorithms SOIL (Gangwani & Peng, 2020) and GAIfO (Torabi et al., 2018b).

We demonstrate the experiment results in Fig. 6 (a). Our ASOR algorithm can increase the performance of PPO by a large margin, while SRPO can only make little improvement. This is because the optimal state distribution in different lava world environments will not be the same. SRPO will still blindly consider all relevant states for policy regularization, leading to suboptimal policies. Fig. 6 (b) demonstrates the average reward with each possible position of the bottom lava block. PPO and PPO+SRPO have low performance when the bottom lava block is at Row 5. They mistakenly regard grids at (5,4) and (5,5) as optimal, but ASOR will recognize these grids as non-accessible states. We also demonstrate in Fig. 6 (c) the disagreement in next state predictions used to compute pseudo count. States far from the starting point have higher prediction disagreements and lower pseudo counts.

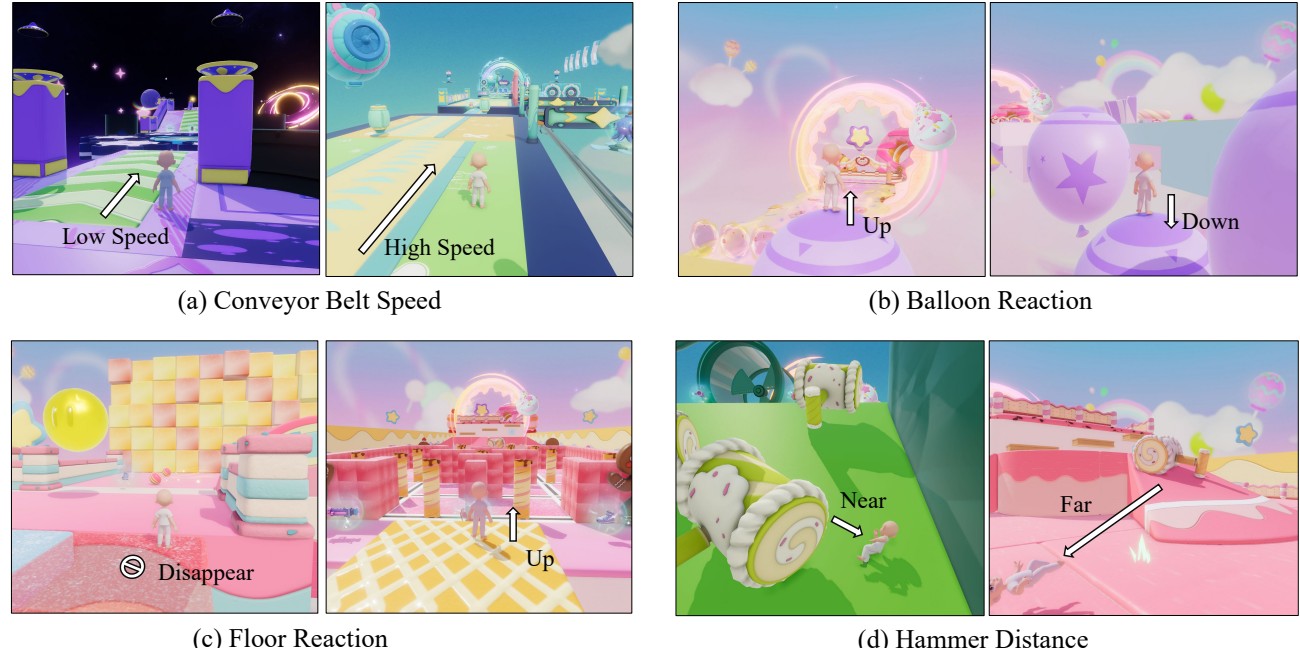

*Figure 7.* More demonstrations of dynamics shifts in fall guys-like game. Colors and textures are only for visual enhancement and are not part of the agent's observations.

### C.3. Additional Setup of the Fall-guys Like Game Environment

**Additional Environment Demonstrations**    Next, we present additional examples of dynamic shifts within the fall-guys like game environment to demonstrate the diverse and variable nature of in-game elements. As shown in Fig. 7, the game environment features a range of dynamic shifts which contribute to the complexity and unpredictability of the gameplay. Specifically, we observe the following scenarios: **Fig. 7 (a)**: The speed of conveyor belts changes across different game settings, leading to varied transitions in the agent's position and momentum when it steps onto these belts. **Fig. 7 (b)**: Balloons exhibit different reactions upon interaction with the agent. This variation can significantly affect the agent's subsequent trajectory. **Fig. 7 (c)**: The behavior of floors under the agent's influence varies significantly. Some floors may collapse, disappear, or shift unexpectedly, introducing further complexity to the environment. **Fig. 7 (d)**: The distance and direction in which the agent is ejected when struck by hammers can vary widely. This variability depends on the unpredictable environmental dynamic shifts, for example, the force and angle of the hammer's swing.

**MDP Setup**    Below, we provide definitions of state space, action space, and rewards in the fall-guys like game environment.

**State space $\mathcal{S}$:**

- **Terrain Map** (dim=$(16 \times 16 \times 2)$ with granularities of $[1.0, 2.0]$): The relative terrain waypoints in the agent's surrounding area. Various granularities capture different details and perceptual ranges effectively.

- **Item Map** (dim=$(16 \times 16 \times 1)$ with granularities of $[1.0, 2.0]$): Map of nearby items or objects. Multiple maps focus on different item types, with granularities for varying spatial scales.

- **Target Map** (dim=$(16 \times 16 \times 1)$ with granularities of $[1.0, 4.0, 16.0]$): Map of archive points locations. Various granularities capture different details and perceptual ranges at different spatial scales.

- **Goal Map** (dim=$(16 \times 16 \times 1)$ with granularities of $[1.0, 4.0, 16.0]$): Map of intermediate goal locations. Various granularities capture different details and perceptual ranges at various spatial scales.

- **Agent Info** (dim=32): Details about agent's own state, including position, rotation, velocity, animation state, and forward direction.

- **Destination Info** (dim=9): Details about the destination, including position, rotation, and size of the destination, providing crucial details for navigation and goal achievement.

**Action space $\mathcal{A}$:**

- **MoveX** (dim=3): Move along the X-axis. The three discrete options typically represent movement in the positive X direction, negative X direction, or no movement.

- **MoveY** (dim=3): Move along the Y-axis, with three discrete options for movement in the positive Y direction, negative Y direction, or no movement.

- **Jump** (dim=2): Represents jump behavior. Options are to initiate a jump or not.

- **Sprint** (dim=2): Represents sprint behavior. Options are to start sprinting or not.

- **Attack** (dim=2): Executes an attack. The two discrete options are to initiate an attack or not.

- **UseProp** (dim=2): Utilizes a prop. The two discrete options indicate whether the prop is used or not.

- **UsePropDir** (dim=8): Determines the direction for prop usage. The eight discrete options offer various directional choices for prop utilization.

- **Idle** (dim=2): Represents idle behavior. Options are to remain idle or not.

**Reward $r$:**

- **Arrive Target** (value=1.0): Rewards the agent for successfully reaching the archive point, with a positive reward of 1.0 upon achievement.

- **Arrive Goal** (value=0.3): Rewards the agent for reaching intermediate goal locations within the environment, with a positive reward of 0.3.

- **Arrive Destination** (value=1.0): Rewards the agent for reaching the final destination or endpoint within the environment, motivating task completion.

- **Goal Distance** (decay rate=0.05): Offers distance-based rewards, varying based on proximity to specific goal locations. Rewards diminish as the agent moves away from the goal, with distinct values for different distance ranges.

- **Fall or Wall** (value=$-1.0$): Penalizes the agent for continuously hitting the wall or falling off a cliff with a penalty of -1.0.

- **Stay** (value=$-0.01$): Penalizes the agent for remaining stationary for extended periods, encouraging continuous exploration and movement.

- **Time** (value=$-0.02$): Penalizes each time step, encouraging efficient decision-making and timely task completion.

**Network Architecture**   The network architecture is structured as follows: The Terrain Map, Item Map, Target Map, and Goal Map are each fed into a convolutional neural network (CNN) with ReLU non-linearity, followed by a fully connected network (FCN). This process yields four separate 32-dimensional vector representations for each respective map. The Destination Info and Agent Info are independently input into attention layers, generating 32-dimensional vectors for each. Subsequently, all 32-dimensional vectors (from the CNNs and attention layers) are concatenated into a single feature vector. The concatenated feature vector undergoes processing by a multi-head FCN to yield various output actions. Additionally, the concatenated feature vector is processed by another FCN to produce a value as the value function estimator.

**Training Setup**   We utilized the Ray RLlib framework (Liang et al., 2018), configuring 100 training workers and 20 evaluation workers. The batch size was set to 1024, with an initial learning rate of $1 \times 10^{-3}$, which linearly decayed to $3 \times 10^{-4}$ over 250 steps. An entropy regularization coefficient of 0.003 was employed to ensure adequate exploration during training. The training was conducted using NVIDIA TESLA V100 GPUs and takes around 20 hours to train 6M steps.

