# OpenReview forum: "Policy Regularization on Globally Accessible States in Cross-Dynamics Reinforcement Learning"
_ICML.cc/2025/Conference — ICML 2025 spotlightposter_

### Official Review · Reviewer_UQdV · 2025-02-25

**Overall Recommendation:** 4

**Summary:**

This paper proposes Accessible State Oriented Policy Regularization (ASOR), a reward shaping technique for offline and online,  off-policy RL with dynamic shift. The work is inspired by the dynamic-agnostic state distribution matching in Imitation from Observation (IfO), but points out that naively imitating expert states could be suboptimal with changed dynamics. Thus, the paper focuses on expert states that are always accessible throughout dynamic changes ("globally accessible"), and tries to maximize reward in RL with a constraint of state occupancy difference ($\mathcal{F}$-distance) between state occupancy on all states and state occupancy on globally accessible states. With Lagrangian derivation, the final implementation is to train a GAN discriminating states with high values or high proxy visitation counts (by RND) with other states, and add a reward shaping term from the GAN onto the original reward. On several offline RL and online RL environments, the proposed method is proved to outperform baselines.

## Update After Rebuttal

I have updated my score accordingly during the author-discussion period; no further update.

**Claims And Evidence:**

Yes in general. This paper made several claims: empirically, the paper claims that matching expert states similar to IfO is a good way to learn cross-dynamics RL, but naively matching expert states without considering accessibility is bad; theoretically, the paper claims that one can convert the problem into a constrained optimization with $\mathcal{F}$-distance limited between state occupancies of the policy and the policy on globally accessible states, and the framework has several theoretical guarantees. With Lagrangian multiplier, the objective can be turned into (as a surrogate) an unconstrained one and optimized with a GAN. I agree with most of the claims, except that:

1. not all IfOs are ignorant to dynamic changes;
 2. there are cases in cross-dynamics RL where globally accessible states are empty sets which this method does not seem to apply.

See "methods and evluation criteria" for details.

**Essential References Not Discussed:**

As mentioned in the "Methods and Evaluation Criteria" section, I feel some IfO works are missing, and the authors should also discuss cross-embodiment works where the globally accessible states are empty sets.

**Experimental Designs Or Analyses:**

Yes, and I found the results to be sufficient to prove the effectiveness of the proposed method. The empirical success and detailed analysis on visual environments Is particularly convincing. However:

1. Even if many methods are tested, I still feel some IfO methods are missing; for example, the IfO papers I mentioned in "Methods and Evaluation Criteria" also considers cross-dynamics generalization.

2. The authors claim in Sec. 4.1 that "IfO approaches have the worst performance because they ignore the reward information ..." . However, there is an easy fix (which is widely used in decision transformer papers [1] as "10%BC") to make use of the reward labels: one can simply select the few trajectories with the highest reward as "expert trajectory", and the rest as "auxiliary unlabeled data" (or simply discard them). There are many IfO papers that can learn from a few expert trajectories with auxiliary unlabeled data, such as SMODICE [2], LobsDICE [3], TAILO [4], MAHALO [5] (which can also do offline RL), etc.

3. Some baselines are a bit out-of-date. For example, CQL is usually considered to be worse than IQL [6], which is a more recognized offline RL algorithm. There are also other newer methods such as XQL [7].

**References**

[1] L. Chen et al. Decision Transformer: Reinforcement Learning via Sequence Modeling. In NeurIPS, 2021.

[2] Y. J. Ma et al. Versatile Offline Imitation from Observations and Examples via Regularized State-Occupancy Matching. In ICML, 2023.

[3] G-H Kim et al. LobsDICE: Offline Learning from Observation via Stationary Distribution Correction Estimation. in NeuriPS, 2023.

[4] K. Yan et al. A Simple Solution for Offline Imitation from Observations and Examples with Possibly Incomplete Trajectories. In NeurIPS, 2023.

[5] A. Li et al. MAHALO: Unifying Offline Reinforcement Learning and Imitation Learning from Observations. In ICML, 2023.

[6] I. Kostrikov et al. Offline Reinforcement Learning with Implicit Q-Learning. In ICLR, 2022.

[7] D. Garg et al. Extreme Q-Learning: MaxEnt RL Without Entropy. In ICLR, 2023.

**Methods And Evaluation Criteria:**

Yes. Overall, the proposed method make sense for the motivation proposed by the paper that one should follow states that are successful and visited by different dynamics. However, there is a concern remaining: The globally accessible state might be an empty set in cross-dynamics RL. This is common in cross-embodiment learning; for example, we want to learn a walking policy for robots with either normal legs or crippled legs, and with different heights. in this case, as the robots may never perfectly achieve the same status, the globally accessible state will become an empty set. Such scenario is also considered in some IfO papers which this paper overlooked [1, 2], and thus the statement "only HIDIL considered state distribution mismatch across different dynamics (in IfO) ..." does not hold.

**References**

[1] Y. J. Ma et al. Versatile Offline Imitation from Observations and Examples via Regularized State-Occupancy Matching. In ICML, 2022.

[2] K. Yan et al. A Simple Solution for Offline Imitation from Observations and Examples with Possibly Incomplete Trajectories. In NeurIPS, 2023.

**Other Comments Or Suggestions:**

1. the title of Sec. 2, "Backgroud" -> "Background";

2. In this paper, $\lambda$ is used as the Lagrange multiplier, regularization coefficient, and Lipschitz coefficient in the same time. This is confusing, as i cannot find the value of $\lambda$ regularization coefficient in this paper (there is an ablation in Tab. 2, but I am unable to find the the value adopted in the main results). The $\lambda$ in Tab. 4 seems to refer to the Lipschitz coefficient.

3. I would suggest the authors to also provide the procedure of ASOR+MAPLE in the appendix besides ASOR+ESCP.

**Other Strengths And Weaknesses:**

**Other Strengths**

1. The paper is overall well-written and easy to follow; the motivation from IfO that "one should focus on expert states, but not all expert states" seems natural.

**Other Weaknesses**

1. The method seems to have many moving parts. For this method, one need to train a RND for proxy of visitation frequency, a GAN, a context encoder (from ESCP), an actor and a critic together. The training stability of this method could potentially be fragile. The authors mention this in Fig. 3, but the augmented reward seems to be very close to 0, which contradicts with the left hand side of Fig. 3 where augmented reward has a very large absolute value. Can the authors explain this?

**Questions For Authors:**

I have a question: how many GPUs does the training use? The authors claim that "The training was conducted using NVIDIA TESLA V100 GPUs and takes around 20 hours to train 6M steps", but did not specify the number of GPUs.

**Relation To Broader Scientific Literature:**

This paper is beneficial to the Reinforcement Learning (RL) / Imitation Learning (IL) community, and potentially helpful for the robotics and computer vision community (as it contains visual environments). It does not affect much to the scientific literature beyond the communities above.

**Theoretical Claims:**

There are two theoretical claims in this paper, which are about finite and infinite sample analysis of the proposed method. I briefly checked both of them and they look sensible and correct to me.

---

> ### Author Rebuttal · Authors · 2025-04-01
>
> We thank the reviewer for the constructive and insightful comments. Responses are provided as follows. The linked file is available [here](https://anonymous.4open.science/api/repo/ICML_ASOR_Rebuttal-01CB/file/rebuttal_append_file.pdf).
>
> **Q1: Possibly empty accessible state set**
>
> A: Tasks with no globally accessible states are indeed extreme cases where ASOR may degrade into the base RL algorithm. But two practical factors can help alleviate such concern.
> - One may regard the state dimension that remains distinct across dynamics as hidden environment parameter and do not include such information when identifying globally accessible states. For example, the robot height in the mentioned example can be excluded in the state space. States that are visited by expert policies with different robot height, e.g., smooth walking with moderate speed, can be regarded as globally accessible and preferred when training new policies.
> - In the practical algorithm procedure of ASOR, a relative accessibility detection mechanism is applied. States that are most likely to be accessible in a training batch are regarded as preferable (Line 7 in Algorithm 1). We can rely on the discriminator network to automatically find appropriate states to assign higher rewards (Line 8 in Algorithm 1).
>
> Meanwhile, the mentioned robotics scenario with different robot leg conditions can also be challenging for base RL algorithms which ASOR is built on, such as cross-domain RL methods. A considerable amount of training data may be needed to train decent policies in such scenarios.
>
> **Q2: Overlooked IfO papers**
>
> A: Thanks for bringing these papers into our attention. We will add discussions in the related work of the revised paper. We also agree with the reviewer that we should give IfO papers [1,2] enough credit for also considering state distribution mismatch. IfO methods are helpful when enough near-expert trajectories or unlabeled data are available and have better training efficiency. The advantage of ASOR lies in that its reward augmentation mechanism can be effectively integrated with both offline and online RL methods and adds minimal changes to their original training pipeline.
>
> **Q3: Additional baselines**
>
> A: Thanks for introducing alternative ways of constructing baselines. We refer the reviewer to Tab. 3 of the linked page, where comparative analysis on SOIL with 10% data and IQL is included. SOIL indeed benefits from the 10% data fix and shows increased performance, while IQL has similar performance with CQL. Both approaches cannot outperform MAPLE+ASOR. This can be because the offline datasets do not have enough state coverage for SOIL to imitate. Meanwhile, IQL and CQL still focus on state-action joint distributions in the offline dataset. Such distribution can be unreliable under dynamics shift since given the same state, the optimal action may be different in different dynamics. We are unable to reimplement the mentioned DICE-related methods due to the limited rebuttal time and will add them in the revision.
>
>
> **Q4: Potentially fragile training stability**
>
> A: Both the RND module and the GAN module are essentially supervised learning on given datasets. Their training will be more stable than the actor-critic module that involves RL loss. The red loss curve in Fig. 3 (right) also indicates that the RND loss and the GAN discriminator loss drop smoothly, indicating a relatively stable training process. One might relate the GAN module to the unstable training of GANs. But in fact, the GAN module only utilizes a GAN-like objective function. Its training procedure is different from GAN in that the data to train the discriminator is directly constructed from the replay buffer, instead of being adversarially generated.
>
>
> **Q5: Contradicting augmented reward**
>
> A: Fig. 3 (left) demonstrates the augmented reward of a single step, while in Fig. 3 (right) we show rewards averaged in a batch and multiplied by the coefficient. The extreme values of the augmented reward only exist when the walker agent is about to fall down near the end of the trajectory, so they only make up a small fraction of a training batch. The average augmented reward will therefore remain relatively stable during training.
>
> **Q6: Repeated $\lambda$**
>
> A: Thanks for pointing out this issue. We will change the Lipschitz coefficient to $\mu$ in the revision. The Lagrange multiplier $\lambda$ is the the same with the coefficient for the augmented reward according to Eq. (5), so we remain the same notation. For the values of $\lambda$ in different experiments, we refer the reviewer to Tab.1 of the linked pdf page.
>
> **Q7: Number of GPUs**
>
> A: We only use one V100 GPU for centralized training in Ray. Distributed environment rollouts are carried out with ~300 CPU threads.

---

> > ### Comment · Reviewer_UQdV · 2025-04-02
> >
> > Thanks for the reviewer's detailed response. I do not have any other concern, and hopefully the authors can modify the paper accordingly. I shall now raise my score from 3 to 4.

---

> > > ### Author Response · Authors · 2025-04-03
> > >
> > > We appreciate the reviewer's prompt and positive response. We thank the reviewer for all the time and efforts in reviewing this paper and will update the paper as required.

---

### Official Review · Reviewer_2QKH · 2025-03-10

**Overall Recommendation:** 4

**Summary:**

The paper addresses the challenge of learning policies when the environment dynamics vary such that expert state trajectories may not always be accessible under dynamics shifts. To overcome this, the authors present the Globally Accessible States with a formal definition and the  F-Distance, a measure of the discrepancy between the accessible state distributions of the current and expert policies. Building on these concepts, they present the ASOR Algorithm, which integrates these ideas as a reward augmentation module. Designed as a plug-and-play component, ASOR can be applied to both online and offline RL approaches.

The method is evaluated across diverse benchmarks, including MuJoCo, MetaDrive, and a large-scale Fall Guys-like environment. Combining theoretical insights with extensive empirical validation, the paper demonstrates that regularizing policies based on accessible states enhances robustness across dynamic shifts.

**Claims And Evidence:**

Most of the paper’s claims are supported by a mix of rigorous theoretical analysis and extensive empirical experiments.

**Essential References Not Discussed:**

The paper builds on SRPO (Xue et al., 2023a) but does not cite works on meta-learning for RL adaptation, such as MAML (Finn et al., 2017), which are highly relevant in learning adaptable policies across diverse dynamics.

**Experimental Designs Or Analyses:**

I examined several aspects of the experimental designs and analyses presented in the paper. Overall, while the chosen benchmarks and metrics are appropriate for cross-dynamics RL, some issues remain on certain experiments. Addressing these problems would strengthen the evaluation of the results:
- Some experimental details, such as the exact procedures for constructing state partitions (DP and DQ) and the calibration of pseudo-counts, are described briefly. This might make it harder for a reader to reproduce the results or fully understand the sensitivity of the approach to these choices.
- Although the authors have provided ablation studies in Table 2, I remain unconvinced of ASOR's robustness to the reward coefficient and state partitioning hyperparameters. First, in the first and third columns, the average result (0.34) is lower than that of the original MAPLE (0.36). The authors should provide more detailed results to illustrate the sensitivity of a broader range of hyperparameter choices. Second, in the final column, it is unclear whether the results across different domains were obtained using the same set of hyperparameters. If not, how were the hyperparameters tuned for each domain? Lastly, perhaps I overlooked it, but I could not find the exact values of the final chosen hyperparameters.
- For the online RL experiments in Figure 2, the authors are encouraged to follow the experimental setups in the SRPO paper more closely, ensuring alignment in iteration numbers and experimental domains. This would help readers better assess whether the proposed method is fairly compared with SRPO. Additionally, in HalfCheetah, ASOR shows only marginal improvements over the baselines. Is there a fundamental reason for its reduced effectiveness in this setting? If the limitation stems from state accessibility not changing significantly in HalfCheetah, this would suggest that ASOR is most beneficial in environments where accessibility varies.
- Regarding the experiments in MetaDrive, the training process appears to be unstable and not fully converged for the compared methods. Could the authors include results with additional training steps for better convergence?
- I find it unclear why the authors describe the Fall Guys-Like Game experiments as large-scale, given that no visual inputs are used and the action space appears to be small. It remains uncertain whether the proposed method would be effective in high-dimensional state and action spaces, such as humanoid or environments with visual observations. Could the authors clarify these points?

**Methods And Evaluation Criteria:**

The proposed methods and evaluation criteria are generally well-aligned with the problem of cross-dynamics reinforcement learning. The core idea of focusing on globally accessible states addresses a real issue in cross-dynamics scenarios. The method directly targets the shortcomings of traditional IfO approaches by regularizing policies to only mimic these reliable states. The theoretical framework (using F-distance with both JS divergence and neural network distance) is solid, even though it relies on certain assumptions that might limit its application in highly adversarial settings.

For the evaluation criteria, the paper evaluates its methods on a variety of benchmark datasets and environments, including MuJoCo, Minigrid, MetaDrive, and a large-scale Fall Guys-like game. These benchmarks cover a range of tasks (both online and offline) and dynamics shifts, providing a comprehensive test bed that is well-suited to assess the proposed method’s performance.

**Other Comments Or Suggestions:**

For the methodology part, the construction of DP and DQ datasets (used for state accessibility filtering) is described briefly but is critical to the method. Adding more details on how the partitioning is implemented and its computational overhead would strengthen reproducibility. Besides, the construction of DP and DQ datasets (used for state accessibility filtering) is described briefly but is critical to the method. Adding more details on how the partitioning is implemented and its computational overhead would strengthen reproducibility.

**Other Strengths And Weaknesses:**

Strengths:
- The paper identifies a critical limitation in many IfO approaches—the assumption of identical expert state distributions across dynamics—which is often violated in real-world scenarios. This insight is both timely and relevant for cross-dynamics RL.
- The derivation of performance bounds using two different instantiations (JS divergence and network distance) provides solid theoretical grounding.
- ASOR is designed as a modular add-on that can be integrated with existing state-of-the-art RL algorithms. Its practical implementation via a GAN-like objective for reward augmentation makes it appealing for both online and offline settings.

Other weaknesses:
- While the paper is well-structured overall, some sections (especially those covering theoretical analyses and the accessible states) could be improved with additional intuitive explanations or visual aids. This would help the readers better understand the paper.

**Questions For Authors:**

1. In Theorem 3.5 and Theorem 3.7, the performance lower bounds rely on the assumption that the HiP-MDP’s MDPs are M-Rs accessible from each other. How restrictive is this assumption in practice? If it fails in many real-world cases (e.g., highly stochastic environments or POMDPs with high-dimensional observations such as visual inputs), then the practical significance of the theoretical guarantees would be weakened.
2. In some cases, the globally accessible states between the current environment and the expert trajectories may not share the same optimal policy. How do the authors account for this situation? Could ASOR have a negative effect in such cases?
3. How stable is the discriminator training for F-distance estimation? Are there cases where the optimization struggles (e.g., mode collapse, vanishing gradients)? Since GAN training can be unstable, an analysis of failure cases or stability techniques (e.g., spectral normalization, gradient penalties) would be valuable. If the discriminator fails in some environments, addressing this limitation (or providing mitigation strategies) would be important for practical applications.
4. Please see our questions in the "Experimental Designs Or Analyses" section above.

**Relation To Broader Scientific Literature:**

The key contributions of the paper relate to multiple areas within reinforcement learning (RL), particularly in cross-dynamics policy transfer, imitation learning from observation (IfO), and policy regularization. For cross-dynamics RL, it extends SRPO by redefining state accessibility and improving transfer in dynamic environments.

**Theoretical Claims:**

I examined several of the theoretical proofs provided in the paper, particularly those supporting the performance lower bounds (Theorem 3.5 and Theorem 3.7).

---

> ### Author Rebuttal · Authors · 2025-04-01
>
> We thank the reviewer for the constructive and insightful comments. Responses are provided as follows. The linked file is available [here](https://anonymous.4open.science/api/repo/ICML_ASOR_Rebuttal-01CB/file/rebuttal_append_file.pdf).
>
> **Q1: Robustness to hyperparams**
>
> A: Thanks for mentioning the hyperparameter table which is indeed missing in the original paper. We refer the reviewer to Tab. 1 and Tab. 2 in the linked file for detailed hyperparameters. In offline RL, we tried two values of $\lambda$: 0.1 and 0.3. For datasets with higher level of optimality, $\lambda=0.1$ is preferred so that less policy regularization is exerted. $\lambda=0.3$ is preferred on datasets with higher randomness to add more policy regularization based on optimal accessible state distributions. $\rho_1$ and $\rho_2$ can be similarly chosen based on offline data optimality with possible values (0.5, 0.5) and (0.3, 0.3). In medium-expert datasets, states are more likely to be sampled from the optimal accessible state distribution, so higher values $\rho_1$, $\rho_2$ are chosen. In online RL, $\lambda$ is set to 0.03 in MuJoCo environments and 0.1 in the Fall Guys-like environment, mainly to fit the reward scale of different RL environments. $\rho_1$, $\rho_2$ are kept to 0.5 in online RL experiments.
>
> In Tab. 2, a fixed $\lambda=0.1$ has poor performance compared with MAPLE. This is because the augmented reward adds additional variance to environment reward. In some datasets its scale may not be large enough to exert reasonable policy regularization, so the reward variance dominates the training process and leads to a performance drop. Meanwhile, random partition denotes that the real data and fake data to train the GAN-like discriminator in Algorithm 1 are randomly chosen from the training batch. Its poor performance compared with MAPLE demonstrates the importance of constructing discriminator training data according to the accessible state distributions.
>
> In Tab. 4 of the linked file, we conduct additional hyperparameter tuning where $\lambda$ ranges from 0.1 to 0.4 and $\rho_1,\rho_2$ equal to 0.3, 0.5, and 0.7. The results demonstrate that ASOR's performance is relatively stable with $\lambda=0.2, 0.3, 0.4$. $\rho_1,\rho_2=0.7$ may include too many states in the real dataset for discriminator training, so the performance drops accordingly.
>
> **Q2: Marginal improvements in HalfCheetah**
>
> A: We agree with the reviewer that ASOR is most beneficial in environments where accessibility varies, such as the fall-guys like game environment. As discussed in Sec. 4.2 (Lines 380~406), the agent will not fall over in the HalfCheetah environment, so the state accessibility will be more likely to remain unchanged under dynamics shift. This undermines the effectiveness of ASOR and leads to the marginal performance improvements.
>
> **Q3: More training steps in MetaDrive**
>
> A: We clip at 1M episodes to make training steps consistent with other environments. We will add the full training log in the revision.
>
> **Q4: The scale of fall guys-like game environment**
>
> A: The fall guys-like environment has a 1065-dimensional state space containing terrain, map, target, goal, agent, and destination information. Although there are no visual inputs, the state space contains diversified information that is challenging to integrate. Meanwhile, due to the highly dynamic environment and complex interactions between the agent and the environment components, a large number of environment steps is needed to train a decent policy. That is why we call the fall guys-like environment "large scale". The scale of the action space will not influence the effectiveness of ASOR, as the proposed policy regularization objective relies solely on the state distribution and do not take the action or policy distribution into account.
>
> **Q5: Restrictiveness of the assumption**
>
> A: As discussed after Def. 3.3 (Lines 208~219), the assumption is weaker than those in existing methods and less restrictive. It requires state $s'$ can be accessed from $s$ after dynamics shift, as long as it is accessible from $s$ in the original dynamics. It does not require a specific policy to do so and does not limit the number of intermediate states. Such assumption still holds in two of the mentioned extreme cases in that the state accessibility does not change.
>
> **Q6: Different optimal policy**
>
> A: According to Eq. (2,3), regularized policy optimization achieves performance lower-bounds without assumptions on how close the optimal policies are. Intuitively, the regularization provides a reasonable starting point for policies to explore more efficiently, but the starting point is not necessarily optimal. The walker2d environment  are known to have many different near-optimal policies, while our ASOR algorithm can still achieve the best performance (Fig. 2).
>
> **Q7: Stability of GAN training**
>
> A: Due to the limited word count, we refer the reviewer to Q4 in the rebuttal to Reviewer UQdV.

---

> > ### Comment · Reviewer_2QKH · 2025-04-02
> >
> > The authors' response has largely addressed my concerns. I have raised my rating by one point.

---

> > > ### Author Response · Authors · 2025-04-02
> > >
> > > We appreciate the reviewer's prompt and positive response. We thank the reviewer for all the time and efforts in reviewing this paper and will update the paper as required.

---

### Official Review · Reviewer_ZkD1 · 2025-03-13

**Overall Recommendation:** 4

**Summary:**

The paper pinpoints a flaw in existing IfO methods where state inaccessibility due to changing environment dynamics can disrupt the similarity of expert state distributions. To tackle this, it presents a policy regularization approach centered on globally accessible states. The proposed framework combines reward maximization and IfO via F-distance regularized policy optimization, leading to the development of the ASOR algorithm through different F-distance instantiations. Experiments across multiple benchmarks demonstrate its effectiveness in enhancing cross-domain policy transfer algorithms, outperforming baselines, and theoretical analysis offers performance guarantees for policy regularization during dynamics shift.

**Claims And Evidence:**

Yes

**Essential References Not Discussed:**

No

**Ethical Review Flag:**

Flag this paper for an ethics review.

**Experimental Designs Or Analyses:**

The experimental designs are sound. In offline RL benchmarks, the authors collect static datasets from environments with different dynamics and compare ASOR with multiple baseline algorithms. In online continuous control tasks, they explore different sources of dynamics shift. In the fall guys-like game environment, they consider highly dynamic and competitive scenarios. The ablation studies in the MuJoCo tasks help to understand the role of different components and hyperparameters in ASOR.

**Methods And Evaluation Criteria:**

Yes

**Other Comments Or Suggestions:**

No.

**Other Strengths And Weaknesses:**

### Strengths:
- The paper is well-written and the concepts are clearly explained.
- Policy regularization on globally accessible states is a novel way to handle dynamics shift in RL.

### Weaknesses:
- Although the definition of accessible states is clear in theory, it may be extremely difficult to accurately determine whether a state is globally accessible in a real-world environment. Real systems may have noise, unmodeled factors, or overly complex dynamic changes, making it hard to determine whether there exists a policy that can access the state with a non-zero probability for all values of the dynamic parameters.
- Sampling from and estimating distributions from different distributions are crucial steps in the algorithm. However, in practice, accurately sampling from these distributions can be challenging, especially sampling from $d^{*, +}_{T_0}(\cdot)$. If the sampling is inaccurate, it may lead to a deviation in the estimation of the state distribution, thereby affecting the training of the discriminator and the learning effect of the policy.

**Questions For Authors:**

No.

**Relation To Broader Scientific Literature:**

The paper builds on the existing literature of cross-domain policy transfer and Imitation Learning from Observation. It addresses the limitations of previous IfO methods that assume similar state distributions across different dynamics.

The paper defines globally accessible states, providing a new way of looking at state spaces in the context of learning from diverse dynamics.

**Theoretical Claims:**

Yes.  For example, Theorem 3.5 shows the lower bound of the learning policy's performance when using JS divergence for F-distance regularization.

---

> ### Author Rebuttal · Authors · 2025-04-01
>
> We thank the reviewer for the constructive and insightful comments. Responses are provided as follows.
>
> **Q1: Determining accessibility in practice**
>
> A: Tasks with potentially empty globally accessible state set are indeed extreme cases where ASOR may degrade into the base RL algorithm. An example proposed by Reviewer UQdV is when we want to learn a walking policy for robots with either normal legs or crippled legs, and with different heights. In this case, as the robots may never perfectly achieve the same status, the globally accessible state will become an empty set. But two practical factors can help alleviate such concern.
> - One may regard the state dimension that remains distinct across dynamics as hidden environment parameter and do not include such information when identifying globally accessible states. For example, the robot height in the mentioned example can be excluded in the state space. States that are visited by expert policies with different robot height, e.g., smooth walking with moderate speed, can be regarded as globally accessible and preferred when training new policies.
> - In the practical algorithm procedure of ASOR, a relative accessibility detection mechanism is applied. States that are most likely to be accessible in a training batch are regarded as preferable (Line 7 in Algorithm 1). Even though there are no states that are perfectly accessible across dynamics, we can rely on the discriminator network to automatically find the most appropriate states to assign higher rewards (Line 8 in Algorithm 1).
>
> **Q2: Sampling from the accessible distribution**
>
> A: According to Sec. 3.5 of the paper, instead of directly sampling from the intractable accessible state distribution $d_{T_0}^{\*,+}(s)$ , we estimate the its likelihood ratio with $d_T^\pi(s)$. The key property used here is Prop. 3.8, which shows that the likelihood ratio can be obtained by optimizing a classifier similar to the GAN discriminator, as long as the real data are sampled from the state distribution in the molecular of the ratio and the fake data from the denominator. According to Eq. (4), the likelihood ratio $\frac{d_{T_0}^{\*,+}(s)}{d_T^\pi(s)}$ can be decomposed into the production of state optimality and accessibility. We therefore use the state value function and state visitation pseudo-count approaches to split the training batch and create classifier training data. In this way, we manage to obtain an estimation of the likelihood ratio, which serves as the augmented reward in practice.

---

### Official Review · Reviewer_yDbB · 2025-03-15

**Overall Recommendation:** 4

**Summary:**

This paper studies policy learning in cases where the environment dynamics may vary during training. Even when expert demonstrations or replay buffers are provided, some states within them may be inaccessible. In such situations, the policy should identify which states are globally accessible and consider only the information from them.

To address this, the paper proposes ASOR, an add-on GAN-like module that identifies states with high values or proxy visitation counts. ASOR is integrated with existing RL methods and achieves superior or competitive performance compared to baselines on various benchmarks, including Minigrid, D4RL, MuJoCo, and Fall Guys-like battle royale games.

**Claims And Evidence:**

This paper highlights the potential impact of changes in the environment's dynamics and their consequences—specifically, the accessibility of states must be considered when learning from collected transitions. Otherwise, the policy may be misled, as it was trained under different dynamics and scenarios.

[+] In my opinion, the claim appears sound, and the theoretical support—ranging from infinite samples and finite samples to a practical GAN-like implementation—seems accurate. Moreover, the paper conducts comprehensive experiments to empirically validate its claims.

[-] However, one concern I have is that even if we determine that a given task and environment can be suitably defined in the form of HiP-MDP, how can we obtain the required information in practice? For example, how do we estimate the accessible state distribution? How many orders of magnitude of transitions are required to obtain a reliable estimate? What should be done if there is a safety issue?

**Essential References Not Discussed:**

[+] Since there is little literature related to HiP-MDP, I believe this study sufficiently cites and discusses the literature in this field.

[-] However, it would make the literature review more complete if HiP-MDP were discussed alongside other potentially related MDP settings, such as LMDP [R1]. Additionally, diffusion and max-ent policies are known to be good at dealing with environmental (dynamics) changes, and I believe they should be discussed and examined in this work.

[R1] Zhou et al., "Horizon-Free and Variance-Dependent Reinforcement Learning for Latent Markov Decision Processes," ICML 2023.

**Experimental Designs Or Analyses:**

[+] ASOR is compared with multiple baseline methods in environments across different fields. Although ASOR achieves sub-optimal results in some cases, I acknowledge that it performs the best overall.

[+] This paper analyzes the experimental results in depth, rather than simply reporting the results of each method, which is commendable.

[-] I believe diffusion and maximum entropy (max-ent) policies should be compared in the experiments, as these policies are effective at handling sudden changes in the environment. Comparing with them and still achieving the best performance would make ASOR's effectiveness and superiority even more convincing.

[-] Another experiment worth exploring is the scenario when the dynamics during training and testing deviate. Since in Algorithm 1, the dynamics are randomly sampled, it may happen that a particular dynamic is rarely sampled, and thus, the policy may not learn well from transitions of that dynamic. One possibility is that a dynamic is excluded during training but may be sampled during inference. In this case, what would be the difference in performance between each method?

**Methods And Evaluation Criteria:**

The ASOR method has the following major components or strategies, including a GAN-like training method, tracking state values and proxy visitation counts, and using a Lagrangian multiplier to augment rewards with the discriminator's output.

[+] These designs make sense to me. Intuitively, they would make the policy more cautious when visiting areas that are likely to change over time. These designs are also supported by the ablation studies (Table 2, Figure 3).

[+] The results for each environment are tested over multiple rounds, and the standard deviation of the method's performance is reported as well.

[-] One question I have: while ASOR may encourage the policy to focus on areas that are not affected by changes in dynamics and have high values, how can it determine which dynamics it is in during inference? For example, taking the lava scenario in Figure 1, the policy may know that there is uncertainty in the intersection areas of rows 2, 3, and 4 and columns 4, 5, and 6. But when it stands at (2, 3), how does it know which situation it is facing now?

**Other Comments Or Suggestions:**

All my comments and suggestions are listed in the appropriate fields above. Since I have rarely engaged in HiP-MDP research before, I am giving a relatively conservative initial recommendation of 'weak acceptance.' However, I am happy to raise my recommendation if my concerns are addressed during the discussion phase and no significant or widely shared concerns are raised by other reviewers.

**Other Strengths And Weaknesses:**

[+] The proof, descriptions of the baseline method, and details of the evaluation environments provided in the appendix allow readers to more fully examine how the study was conducted and provide sufficient reproducibility.

[-] A minor typo is found at line 383: "ASOR's will have ...".

**Questions For Authors:**

My concerns and questions have been summarized in the above fields, please consider addressing those points marked with [-].

**Relation To Broader Scientific Literature:**

[+] Although HiP-MDP is a relatively new and niche research direction, I find it more aligned with practical scenarios and believe it has the potential to enhance RL applications in real-world settings. I appreciate efforts to explore this direction and propose novel methods to push its boundaries.

**Theoretical Claims:**

[+] As mentioned above, the theoretical claims are sound, and their proof seems accurate. Those toy examples and illustrations play a critical role, helping the reader follow the derivation even though it is somewhat cumbersome.

[-] Some notations use superscripts and subscripts to represent different meanings, which can easily be confused while reading, such as $d_{T}^{\pi^{*}}(s)$ , $d_{T}^{\*}(s)$, $d_{T_{\theta}}^{\pi}(s)$, and $d^{\pi, +}_{T}(s)$. If a clearer and more concise way of expression exists, the theoretical paragraphs will be easier to read.

---

> ### Author Rebuttal · Authors · 2025-04-01
>
> We thank the reviewer for the constructive and insightful comments. Responses are provided as follows. The linked file is available [here](https://anonymous.4open.science/api/repo/ICML_ASOR_Rebuttal-01CB/file/rebuttal_append_file.pdf).
>
> **Q1: Practical estimation concerns**
>
> A: We discuss in Sec. 3.5 on how to obtain practical estimations. Instead of directly estimating the intractable accessible state distribution $d\_{T_0}^{\*,+}(s)$ , we estimate the its likelihood ratio with $d_T^\pi(s)$. The key property used here is Prop. 3.8, which shows that the likelihood ratio can be obtained by optimizing a classifier similar to the GAN discriminator, as long as the real data are sampled from the state distribution in the molecular of the ratio and the fake data from the denominator. According to Eq. (4), the likelihood ratio $\frac{d_{T_0}^{\*,+}(s)}{d_T^\pi(s)}$ can be decomposed into the production of state optimality and accessibility. We therefore use the state value function and state visitation pseudo-count approaches to split the training batch and create classifier training data. In this way, we manage to obtain an estimation of the likelihood ratio, which serves as the augmented reward in practice.
>
> For the amount of transitions required, we refer the reviewer to Fig. (3)(right), where the red line (extra loss) contains the loss of training the estimation network. It takes roughly 1M steps to obtain a decent estimation, which is 1/6 of the total policy training steps. In other words, a relatively accurate likelihood ratio estimation is easier to obtain compared with a good RL policy, and can provide reasonable guidance in the policy training process.
>
> When there are safety issues, safe RL techniques can be integrated as ASOR does not change the training pipeline of the base algorithms. One may also increase the coefficient of the augmented reward because it encourages the policy to stick to states that are more likely to be visited by expert policies in different dynamics.
>
> **Q2: Dynamics identification**
>
> A: Different base algorithms have their own approach of identifying environment dynamics, and ASOR as a general reward augmentation algorithm is agnostic to these approaches. For example, MAPLE and ESCP use context encoders trained with auxiliary loss to detect dynamics change. In the fall-guys like game environment, we include transformers in the policy network to automatically determine environment dynamics from history state-actions. In the lava scenario, we add environment information in the state space for simplicity. Apart from location coordinates, the state space has a 0-1 variable indicating whether there is a lava block nearby (Lines 153-158). In Fig. 1 (up), the agent at (2,3) will have (2,3,1) as the state; In Fig. 1 (bottom), the state will be (2,3,0). The agent can therefore be informed which situation it is facing.
>
> We will add the aforementioned discussion to the revision.
>
> **Q3: Confusing notations**
>
> A: Sorry for the confusing notations.  $d_T^*(s)$ is the brief version of $d_T^{\pi^*}(s)$ and means the same distribution. For the accessible state distribution such as $d^{\pi,+}_T(s)$, we plan to use a new distribution notation $\delta^\pi_T(s)$ for better readability.
>
> **Q4: Experimental designs**
>
> A: Thanks for recommending diffusion policies as baseline. We consider the Diffusion-QL algorithm and refer the reviewer to Tab. 3 of the linked pdf for comparative analysis. It exhibits better performance than CQL especially in the Hopper environment, but cannot outperform the average performance of MAPLE+ASOR. This can be because Diffusion-QL tries to recover the state-action joint distribution in the offline dataset. Such distribution will be unreliable under dynamics shift since given the same state, the optimal action may be different in different dynamics. For maximum entropy policies, we included SAC as baseline in online RL experiments. Meanwhile, ESCP and ESCP+ASOR are built upon SAC themselves and will benefit from the maximum entropy training objective.
>
> For experiments with deviated dynamics during training and testing, i.e., testing with out-of-distribution dynamics, we refer the reviewer to Fig. 1 of the linked pdf. We utilize the OOD setting in ESCP, where test environment parameters, including the damping coefficient and the wind speed, are sampled from distributions that are 33% broader than the training distribution. In this scenario, HalfCheetah and Ant witness performance drops across different algorithms.  Algorithms show almost no performance changes in the Walker2d environment on unseen test environments. This may be because dynamics shift exerts less influence on the walker agent. In both scenarios, ESCP+ASOR still achieves the best results, demonstrating its ability to efficiently train a more robust policy.
>
>
> **Q5: Literature completeness**
>
> A: Thanks for bringing LMDP and diffusion RL research into our attention. We will add discussions on these related works.

---

> > ### Comment · Reviewer_yDbB · 2025-04-02
> >
> > I appreciate the authors' effort in addressing my concerns and questions. All my concerns have been well addressed or clarified. After reviewing the discussions in other reviewers' threads, I did not find any major concerns shared among the reviewers. Therefore, I am happy to raise my score to 'Accept' to acknowledge the authors' effort. Great job, and good luck!

---

> > > ### Author Response · Authors · 2025-04-02
> > >
> > > We appreciate the reviewer's prompt and positive response. We thank the reviewer for all the time and efforts in reviewing this paper and will update the paper as required.

---

### Decision · Program_Chairs · 2025-05-01

**Decision:**

Accept (spotlight poster)

**Comment:**

The authors present an approach to learn from environments with varying dynamics by combining reward maximization with imitation from observations by using f-divergence between state distributions for policy optimization. This allows them to only consider globally accessible states, with the assumption that different dynamics across environments can render some states inaccessible. The theoretical framework is a strong contribution, showing how different f-divergences lead to different analyses. The empirical contributions are also strong, evaluating on a variety of benchmark datasets and environments, including MuJoCo, Minigrid, MetaDrive, and a large-scale Fall Guys-like game. These benchmarks cover a range of tasks (both online and offline) and dynamics shifts, providing a comprehensive test bed that is well-suited to assess the proposed method’s performance.